# In Vitro Proof-of-Concept Study: Lidocaine and Epinephrine Co-Loaded in a Mucoadhesive Liquid Crystal Precursor System for Topical Oral Anesthesia

**DOI:** 10.3390/ph18081166

**Published:** 2025-08-06

**Authors:** Giovana Maria Fioramonti Calixto, Aylla Mesquita Pestana, Arthur Antunes Costa Bezerra, Marcela Tavares Luiz, Jonatas Lobato Duarte, Marlus Chorilli, Michelle Franz-Montan

**Affiliations:** 1Department of Biosciences, Piracicaba Dental School, Universidade Estadual de Campinas, Av. Limeira, 901, Piracicaba 13414-903, SP, Brazil; giovana.balian@gmail.com (G.M.F.C.); pestanaaylla@gmail.com (A.M.P.); arthurantunes32@gmail.com (A.A.C.B.); 2Department of Drugs and Medicines, School of Pharmaceutical Sciences, São Paulo State University—UNESP, Araraquara 14800-903, SP, Brazilmarlus.chorilli@unesp.br (M.C.)

**Keywords:** buccal mucosa, drug delivery systems, lamellar crystalline liquid system, local anesthetic, nanotechnology

## Abstract

**Background:** Local anesthesia is essential for most dental procedures, but its parenteral administration is often painful. Topical anesthetics are commonly used to minimize local anesthesia pain; however, commercial formulations fail to fully prevent the discomfort of local anesthetic injection. **Methods:** We developed and characterized a novel lidocaine and epinephrine co-loaded liquid crystalline precursor system (LCPS) for topical anesthesia. The formulation was structurally characterized using polarized light microscopy (PLM) and small-angle X-ray scattering (SAXS). Rheological behavior was assessed through continuous and oscillatory rheological analyses. Texture profile analysis, in vitro mucoadhesive force evaluation, in vitro drug release and permeation studies, and an in vivo toxicity assay using the chicken chorioallantoic membrane (CAM) model were also conducted. **Results:** PLM and SAXS confirmed the transition of the LCPS from a microemulsion to a lamellar liquid crystalline structure upon contact with artificial saliva. This transition enhanced formulation consistency by over 100 times and tripled mucoadhesion strength. The LCPS also provided controlled drug release, reducing permeation flow by 93% compared to the commercial formulation. Importantly, the CAM assay indicated that the LCPS exhibited similar toxicity to the commercial product. **Conclusions:** The developed LCPS demonstrated promising physicochemical and biological properties for topical anesthesia, including enhanced mucoadhesion, controlled drug delivery, and acceptable biocompatibility. These findings support its potential for in vivo application and future clinical use to reduce pain during dental anesthesia procedures.

## 1. Introduction

Local anesthetics are commonly used in dental procedures such as tooth extractions, endodontic treatments, restorative procedures, and periodontal therapies, as well as for managing pain associated with oral conditions like mucositis, aphthous ulcers, and oral tumors [1,2]. These drugs work by reversibly inhibiting the perception of sensations, particularly pain, through the blockade of voltage-regulated sodium channels, which prevents the propagation of nerve impulses [1].

Among the various anesthetics used in dentistry, lidocaine hydrochloride (LH) is the most widely utilized [3], being known for its rapid and effective action and has an extremely low incidence of allergic reactions, which makes it suitable for use in children, the elderly, and pregnant individuals [4]. Despite these advantages, the parenteral administration of LH can cause significant discomfort, especially in patients with needle phobia [5], a condition that affects approximately 10% of the global population and can negatively impact patient compliance and treatment outcomes, thereby compromising both oral and systemic health [6,7].

To mitigate this discomfort, topical anesthesia is often applied before injections [8]. However, topical anesthetics face challenges such as limited drug permeation across the mucosal epithelium and rapid elimination from the application site due to activities like chewing, speaking, swallowing, and continuous saliva washout [5,9]. Therefore, there is a need for new strategies to develop safe, effective, and long-lasting topical anesthetics for dental use.

One promising approach is the combination of LH with a vasoconstrictor agent like epinephrine, which is commonly used in injectable solutions. Vasoconstrictors can reduce the systemic absorption rate of LH, prolong its contact with nerve endings, enhance the duration and intensity of anesthesia, and reduce bleeding during dental procedures [10,11]. The combination of lidocaine and epinephrine is widely used in clinical practice due to its synergistic effects, where epinephrine prolongs the anesthetic effect of lidocaine by vasoconstriction, reducing systemic absorption. This combination is endorsed by regulatory agencies such as the FDA, EMA, and RENAME [12,13,14], and is supported by both foundational pharmacological reviews and recent clinical studies demonstrating its efficacy in dental procedures [15,16].

Another strategy involves the use of nanostructured delivery systems for topical anesthetics, which can prolong drug release at the target site and minimize adverse effects. Liquid crystalline precursor systems (LCPSs) are among the investigated delivery systems [17,18]. The characteristics of LCPSs present several advantages over traditional formulations or other drug delivery systems. Firstly, they provide enhanced drug release by offering a controlled and sustained release of the active ingredient, leading to improved therapeutic efficacy and reduced side effects. This reduction in side effects occurs because LCPSs limit systemic absorption by maintaining the drug at the application site for longer periods. Their mucoadhesive properties also enhance retention on the mucosal surface, allowing for lower drug concentrations to be used effectively, which minimizes exposure to non-target tissues and reduces the risk of systemic toxicity [19]. Secondly, these systems are known for their increased stability due to their thermodynamic properties, allowing them to maintain their structure and performance over various temperatures and storage conditions, potentially offering a longer shelf life. Additionally, these systems enhance the solubility of poorly soluble drugs, which is particularly beneficial for topical formulations, as it ensures higher local drug concentration at the mucosal surface, improving therapeutic efficacy. Finally, they can be formulated into various forms such as gels, creams, or liquids, providing greater flexibility in application and improving patient compliance and comfort [20,21].

These systems, composed of surfactants, oils, and water, can form lamellar, hexagonal, or cubic mesophases depending on their concentration. These mesophases exhibit varying viscosities, with lamellar being the least viscous, hexagonal having intermediate viscosity, and cubic being the most viscous. The choice of mesophase depends on the application area and required viscosity [20,22,23]. These isotropic liquid formulations are easy to administer in the oral cavity and undergo phase transitions upon contact with saliva, forming viscous liquid crystalline mesophases in situ. This viscosity change facilitates both the application and retention of the formulation at the desired site [20,21]. Additionally, LCPSs can incorporate two drugs with different physicochemical properties, such as LH and epinephrine, and include mucoadhesive polymers to enhance contact time with the buccal mucosa [22]. Chitosan, a widely used mucoadhesive polymer, optimizes drug delivery in the buccal mucosa due to its excellent mucoadhesive properties resulting from electrostatic interactions with negatively charged saliva [24].

In this context, the present study aims to develop a mucoadhesive LCPS for the delivery of lidocaine hydrochloride and epinephrine as a strategy to optimize topical anesthesia in the oral mucosa. This study is based solely on in vitro experiments to evaluate the formulation’s mucoadhesion, permeation ability, and toxicity. The findings from these in vitro studies will provide a foundation for future in vivo investigations to fully assess the clinical potential of the developed system.

## 2. Results

### 2.1. Development and Structural Characterization of Formulations

The ternary phase diagram was meticulously constructed by varying the proportions of Procetyl^®^ AWS, oleic acid, and a 0.5% chitosan dispersion, resulting in 54 different formulations. Within this diagram, formulation F was strategically chosen, situated in an area denoting a transparent liquid system and positioned near a transition region from a liquid to a viscous system with an increase in the aqueous phase (Figure 1A). This formulation, identified as F, served as the baseline for subsequent investigations.

To mimic the conditions within the oral cavity, in a pilot study, formulation F was diluted with various concentrations of artificial saliva. It was observed that the phase transition of F occurred with the addition of 30% and 100% artificial saliva. These saliva concentrations were similarly identified in previous studies from our group when developing LCPSs for intraoral use [20,25].

Formulation F was then diluted with 30% (*w*/*w*) artificial saliva, resulting in F30, and with 100% (*w*/*w*) artificial saliva, yielding F100. This dilution led to a notable increase in the viscosity of formulation F, thereby altering its visual appearance (Figure 1B). The introduction of lidocaine and epinephrine into formulation F maintained this behavior, as observed in the diluted formulations FL30, FL100, FLE30, and FLE100, all exhibiting higher viscosity compared to their undiluted counterparts FL and FLE.

For comparison, the commercial formulation (FC) of lidocaine 50 mg/g dermatological ointment (orange flavor) displayed an opposing trend (Figure 1C). When artificial saliva was introduced, FC30 and FC100 experienced a drastic decrease in viscosity, ultimately transitioning into a liquid state. The detailed composition of all formulations, including the percentages of surfactant, oil, aqueous phase, active compounds, and artificial saliva, is presented in Appendix A.

Microscopic characterization revealed that formulation F was classified as a microemulsion, evident in the dark field visualized by the polarized light microscope (Figure 1C). This characteristic persisted with the incorporation of lidocaine (FL) and epinephrine (FLE). Photomicrographs further revealed that after dilution of F with 30% artificial saliva, F30 adopted a hexagonal liquid crystalline system, indicated by the visualization of striated structures. In contrast, F diluted with 100% artificial saliva (F100) exhibited a cubic liquid crystalline system.

However, the microscopic structure of F30 underwent alterations after the integration of LH and epinephrine. Photomicrographs of FL30 and FLE30 showcased Maltese cross-shaped structures, leading to their classification as lamellar crystalline liquid systems (Figure 1C). Remarkably, the presence of LH and epinephrine did not induce changes in the microstructure of F100, affirming that FL100 and FLE100 retained their classification as cubic liquid crystalline systems.

To confirm the liquid crystal structures identified, Small-Angle X-ray Scattering (SAXS) was employed as a complementary technique. This approach aimed to validate the outcomes derived from polarized light microscopy (PLM), ensuring a more precise understanding of the nanostructure within the formulations. SAXS measurements provided enhanced insights into the nanostructure, corroborating and expanding upon the information obtained through PLM assays.

The SAXS curves of formulations F, F, FL, and FLE revealed broad peaks, indicative of micellar or microemulsified systems (Figure 2). This observation aligned seamlessly with the findings from PLM, thereby reinforcing the conclusions drawn from the photomicrographs.

Notably, FLE30 exhibited a distinctive correlation in its SAXS data, where the distances of the scattering objects (d_1_/d_2_ = 2 and d_1_/d_3_ = 3) suggested a periodicity equivalent to a liquid crystalline arrangement of a lamellar phase (Appendix A). This finding was consistent with the presence of “Maltese crosses” observed in the PLM assay, confirming the liquid crystalline nature of the formulation.

Further exploration revealed that F30, FLE30, and FLE100 displayed three SAXS peaks with q values in the ratios of 1:2:3, characteristic of lamellar mesophases. FL100, although exhibiting only two peaks with a d_1_/d_2_ ratio of 2, also suggests a lamellar arrangement. In PLM, F30 and FLE30 exhibited Maltese crosses, confirming the presence of lamellar phases. In contrast, FL100 and FLE100 were classified as cubic under PLM, despite their SAXS profiles suggesting lamellar characteristics. This divergence may indicate transitional or coexisting mesophases. The presence of striated textures or dark fields in PLM further supports the notion that systems with liquid-crystalline arrangements are structurally complex, requiring the integration of complementary techniques for accurate characterization. This interplay between SAXS and PLM highlights the nuanced behavior of these systems and underscores the importance of a multifaceted analytical approach, as also discussed by Stribeck in the context of lamellar two-phase systems with structural heterogeneity and stacking disorder [26].

Rheological characterization played a key role in unraveling the dynamic behavior of the formulations under study. Notably, the commercial formulations FC30 and FC100 exhibited flow rheological curves indicative of Newtonian flow (Figure 3A), where viscosity (η) remained constant (η = 1.0). In contrast, the other formulations displayed pseudoplastic characteristics (η < 1.0), signifying a reduction in viscosity under shear stress.

A distinctive feature observed across all formulations was the non-superimposition of ascending and descending rheological curves, creating a region of hysteresis—a characteristic hallmark of thixotropy (Figure 3B–D). This behavior underscored the formulations’ ability to undergo reversible changes in viscosity, adapting to shear stress and recovering their original state when the stress was removed.

Furthermore, the impact of dilution with 100% artificial saliva on rheological properties was investigated. The consistency index, yield strength, and thixotropy of formulations F, FL, and FLE exhibited a significant increase (*p* < 0.0001) upon dilution, as detailed in Appendix A. This finding indicated that the addition of saliva led to a notable enhancement in the structural integrity and thixotropic behavior of these formulations.

Conversely, the commercial formulation (FC) responded differently to dilution, showing a significant decrease (*p* < 0.0001) in the consistency index, yield strength, and thixotropy with the addition of 30% and 100% artificial saliva (Appendix A). Among the formulations diluted with 100% saliva, FL100 demonstrated the highest values for all rheological parameters, while FC100 exhibited the lowest values. This divergence in behavior highlights the distinct rheological responses of the formulations to saliva dilution, emphasizing the need to consider formulation-specific characteristics in oral drug delivery systems.

The temporal evolution of the storage (G’) and loss (G”) moduli as a function of the applied frequency was also studied (Figure 4). Qualitative analysis of the graphs indicated that formulations FC, F, F30, F100, FL100, FLE30, and FLE100 exhibited elastic characteristics (G’ > G”), while FC30, FC100, FL, and FLE presented viscous characteristics (G” > G’).

Both the formulation and the artificial saliva and the interaction of the two variables have a highly significant effect on all continuous rheological parameters (*p* < 0.0001, two-way ANOVA) (Appendix A).

Dilution with 30% and 100% artificial saliva resulted in an increase in the mean values of the G’ modulus and gel strength for formulations F, FL, and FLE, with a significant increase (*p* < 0.0001) observed at 100% dilution for all these formulations. Conversely, dilution of the commercial formulation (FC) with 30% and 100% artificial saliva led to a decrease in the mean values of the G’ modulus and gel strength, and an increase in the loss tangent, with no significant difference between the diluted formulations (FC30 and FC100) (Appendix A). For all test formulations, increasing the concentration of artificial saliva decreased the viscoelastic exponent, whereas for the commercial formulation, it increased.

The mechanical properties of the formulations were meticulously examined through texture profile analysis (Table 1). Interestingly, the commercial formulation exhibited higher average values for mechanical properties compared to all the test formulations. However, a noteworthy transformation occurred with the dilution of FC30 and FC100 with 30% and 100% saliva, rendering them into a liquid state and consequently resulting in the absence of hardness, compressibility, adhesion, and cohesion values. In contrast, formulations F, FL, and FLE, initially characterized by low viscosity, did not exhibit hardness, compressibility, adhesion, or cohesion values. Upon dilution with 30% saliva, these formulations displayed discernible mechanical parameters, with an overall increase observed as the concentration of artificial saliva rose. This observation highlights the dynamic influence of saliva dilution on the mechanical properties of these formulations, particularly when transitioning from a more viscous to a liquid state.

The introduction of lidocaine hydrochloride and epinephrine into the formulations resulted in a decrease in the mean values of mechanical properties compared to the drug-free formulation. However, statistical analysis indicated that the differences between the means of FL100 and FLE100, as well as FL30 and FLE30, were not significant. This finding suggests that the addition of epinephrine did not substantially alter the mechanical properties compared to the formulation with local anesthetic alone. The nuanced impact of these active ingredients on mechanical properties underscores the need for a comprehensive understanding of how drug components interact within the formulations, providing valuable insights for the development of buccal drug delivery systems.

The evaluation of mucoadhesion provided valuable insights into the adhesive properties of the formulations. An interesting observation emerged: the addition of saliva resulted in a decrease in the mean values of peak mucoadhesive strength for all formulations, except for F, which maintained its value. This phenomenon highlights the complex interplay between the formulations and saliva, suggesting that the presence of saliva could potentially compromise the mucoadhesive strength of most formulations.

Intriguingly, the introduction of 100% saliva had divergent effects on the work of mucoadhesive strength. For FL and FLE, there was an increase in the work of mucoadhesive strength, indicating more sustained adhesion, while for FC, the work of mucoadhesive strength decreased. This contrast suggests that the addition of 100% saliva can have formulation-specific impacts on the dynamic aspects of mucoadhesion.

Among the formulations diluted with 100% saliva, FLE100 stood out with a mucoadhesive strength approximately three times higher than FC100. This substantial difference underscores the unique mucoadhesive properties conferred by the incorporation of both lidocaine hydrochloride and epinephrine in the formulation. The enhanced mucoadhesive strength of FLE100 compared to FC100 suggests that the combination of lidocaine and epinephrine may synergistically enhance the adhesive interactions between the formulation and the mucosal surfaces.

These findings highlight the importance of considering not only the presence of saliva but also the specific formulation components when evaluating mucoadhesive characteristics in the development of oral drug delivery systems.

### 2.2. In Vitro Release Study

At the outset of this study, the lidocaine hydrochloride content in the F samples was confirmed to be 100%. The release profiles of local anesthetic from both the FC and FL formulations were evaluated over time (Figure 5). During the initial 6 h, lidocaine hydrochloride release from the FC and FL formulations reached 35% and 4%, respectively, indicating a relatively higher and faster release from the FC in the early stages of the release study. Over the full 24 h period, the maximum lidocaine hydrochloride release observed was approximately 40% for FC and 8% for FL.

The kinetics of lidocaine release from FL and FC were elucidated using mathematical modeling, employing various models such as zero order, first order, Korsmeyer-Peppas, Higuchi, and Weibull (Table 2).

Evaluation based on the coefficient of determination (R^2^) values revealed that, for FL and FC, the Weibull model (case 3—highlighted in bold letter) provided the best description of lidocaine hydrochloride released. Notably, the value of the shape parameter (b) for both formulations was less than 1, indicating that lidocaine release followed Fickian diffusion, which postulates that the flow of the drug goes from regions of high concentration to regions of low concentration, with a magnitude proportional to the concentration gradient. These values are consistent with case 3 which exhibits a higher initial slope followed by an exponential curvature, as is evident from the release profiles for the formulations.

### 2.3. In Vitro Permeation Study

The permeation profiles of lidocaine hydrochloride through porcine buccal mucosa from both FC and FL formulations were systematically investigated in accordance with the dosing guidelines outlined in the package insert of the commercial formulation, which is approved by the Brazilian Health Regulatory Agency (ANVISA). According to these guidelines, the onset of action for a dose of 1 g to 5 g of lidocaine ointment on the patient’s oral mucosa occurs within 0.5 to 5 min post application, with significant pain reduction observed within the first 3 h; however, the analgesic effect diminishes after 3 h. In our study, we opted the worst-case scenario in terms of lidocaine dose and duration of action, applying approximately 1 g of the formulation over a 3 h period.

Figure 6 illustrates cumulative amount of lidocaine permeated over time (µg/cm^2^) across swine buccal mucosa from both the commercial formulation and the liquid crystal precursor system, each containing 5% lidocaine hydrochloride. Linear regression analysis of the permeation curves indicated significantly higher permeation in the commercial formulation (*p* < 0.0001).

The calculated permeation kinetic parameters are presented in Table 3.

The steady-state flux of the commercial formulation FC was higher (*p* < 0.05) than that of the developed formulation FL. This is expected since FL proved to be more structured than FC, as indicated by the rheological, structural, and mechanical properties, which can trap the drug and release it more slowly for permeation. On the other hand, concerning lag time, FL showed immediate permeation, resulting in a faster onset of action.

### 2.4. In Vivo Chicken Chorioallantoic Membrane (CAM) Assay

The toxicity classification of F, FL, FLE, FC, the positive control (sodium hydroxide at 0.1 mol/L) and the negative control (0.9% saline solution) is shown in Table 4. Sodium hydroxide (NaOH) at 0.1 mol/L and 0.9% saline solution were selected as the positive and negative controls, respectively, following HET-CAM Protocol No 96 [27,28]. NaOH is a well-known strong alkaline substance that induces significant cellular damage, whereas saline solution is isotonic and isosmotic, making it a non-irritant to the CAM, as widely documented in previous studies [29,30,31].

The vascular changes before (T0) and after 5 min of administration (T5) of F, FL, FLE, FC, positive control, and negative control are shown in Figure 7.

As shown in Table 5 the formulations F, FL, and FLE caused a slight-to-moderate irritant effect on the CAM, similar to the commercial lidocaine formulation (FC). The presence of lidocaine in the LCPS formulation (FL) resulted in the same toxicity classification (slight to moderate irritant) as the plain LCPS (F) (moderate irritant). Conversely, the inclusion of epinephrine (FLE) reduced the toxicity classification from moderate to slight irritant, making it less toxic than both FL and FC (commercial lidocaine) which were both classified as moderate irritant in this in vivo toxicity model.

## 3. Discussion

In this proof-of-concept in vitro study, we successfully developed and evaluated a novel mucoadhesive liquid crystal precursor system co-loaded with lidocaine and epinephrine, designed for topical application on the oral mucosa. The development of the LCPS demonstrated several advantageous characteristics for buccal drug delivery in the presence of saliva, including enhanced consistency, improved mucoadhesion, sustained release and permeation, and reduced toxicity compared to the commercial formulation, suggesting its potential clinical safety. A comparative summary of the key physicochemical and functional properties of the developed formulation versus the commercial control is provided in Appendix A.

To support the selection of the LCPS, the ternary phase diagram used to develop the formulations was constructed based on systematic variation in surfactant, oil, and aqueous phase ratios, and the results were reproducible across multiple batches. The LCPS was chosen from a region of the diagram that consistently produced transparent, low-viscosity systems with favorable transition behavior upon dilution with artificial saliva. This formulation demonstrated optimal characteristics in terms of drug incorporation, mucoadhesion, and rheological behavior. Although long-term stability and large-scale production were not the focus of this proof-of-concept study, the components used are pharmaceutically accepted and scalable. Future work will include stability studies under different storage conditions and process validation to ensure reproducibility and feasibility for industrial-scale manufacturing.

Considering the wet environment of the oral mucosa, attributed to the presence of saliva covering its entire surface and its negatively charged composition (water, inorganic salts, lipids, and glycoproteins) [32], the development of LCPSs based on mucoadhesive cationic polymers emerges as a promising strategy. These systems are liquid microemulsions that transform into viscous and mucoadhesive liquid crystals upon contact with saliva [20]. In this study, we developed a microemulsion composed of Procetyl^®^ AWS, oleic acid, and chitosan, which transitioned into a cubic-phase liquid crystal (an isotropic gel) when mixed with saliva. This composition was designed based on the ability of oleic acid to act as a topical absorption enhancer [33,34], the biocompatible properties of chitosan, particularly its strong mucoadhesive capacity due to electrostatic interactions between its cationic nature and the negatively charged saliva [35], and the swelling properties of Procetyl^®^ (a biocompatible surfactant) upon contact with water [36]. Procetyl^®^ AWS (PPG-5-CETETH-20) has an amphiphilic structure, which enables the formation of stable liquid crystalline phases upon contact with aqueous environments. It is biocompatible and enhances both drug solubilization and mucoadhesion, making it particularly suitable for mucosal applications [33]. Thus, the combination of these components resulted in improved mucoadhesion and an optimal consistency for topical application on the oral mucosa. Our results indicate that the developed formulation holds promise for future in vivo studies, as it provides a sustained-release system with an absorption promoter and high mucoadhesive capacity [5].

This study successfully incorporated lidocaine hydrochloride, the gold standard anesthetic in dentistry, [37] along with the vasoconstrictor epinephrine to enhance the duration and intensity of lidocaine anesthesia and control minor superficial bleeding [38]. Therefore, the developed formulation is expected to provide more effective and long-lasting topical anesthesia compared to commercial formulations. However, a limitation of this study is the lack of an efficacy assessment of the developed formulation. Future studies will be conducted to confirm its anesthetic performance and clinical applicability.

Importantly, epinephrine is chemically unstable in aqueous solutions due to its catechol structure, which is highly prone to oxidation [39]. Exposure to light, oxygen, or elevated pH can lead to degradation and formation of colored byproducts. In this study, no antioxidants or stabilizers were used, as all LCPS formulations were freshly prepared and handled under light-protected conditions to minimize degradation. Nonetheless, for clinical and industrial applications, future studies should address long-term stability and explore the incorporation of stabilizing agents.

The transition from a microemulsion to a cubic structure in the presence of saliva, as observed in this study, can be explained by the critical packing parameter (CPP). The CPP considers surfactant properties such as the area of the polar group, volume, and length of the nonpolar chain, all of which influence the curvature of the polar-nonpolar interface and, consequently, the type of liquid crystal formed [33]. The increased water content from saliva allowed the polar head groups of Procetyl^®^ to move more freely, inducing disorder in the nonpolar chain and increasing the volume of the hydrophobic region. This led to a higher CPP, facilitating the transformation from a microemulsion to liquid crystals [18,25,33], as illustrated in Figure 1.

The addition of hydrophilic drugs, such as lidocaine hydrochloride, can promote the formation of lamellar phases by increasing the polar area of surfactant [18]. However, 100% saliva was added, no differences in the mesophase were observed between the formulations, suggesting that the excess water from saliva had a more significant impact on the CPP value than the presence of lidocaine hydrochloride, as previously demonstrated [40].

The phase behavior of the liquid crystal precursor system is highly relevant for the topical administration of drugs in the oral cavity. Changes in CPP values induced by excess water also lead to rheological modifications in the mesophases, transitioning from a liquid formulation to a viscous formulation in situ [33], as demonstrated in this study. Viscous formulations tend to remain in contact with the mucosa for longer periods, as confirmed in mucoadhesion assays. These rheological changes can also influence the dissolution and release rate of the drug [41], which was further demonstrated in this study.

All developed formulations (F, FL and FLE) exhibited the same rheological behavior, acting as Newtonian fluids with low consistency and flow limit, which facilitates clinical administration. However, when diluted with artificial saliva, in contrast to the commercial formulation that liquefied, they became viscous pseudoplastic formulations with high consistency, flow limit, and thixotropy, desirable characteristics for topical application formulations [42]. These formulations maintain their viscosity at rest, stabilized by intermolecular interactions [43]. When a critical force is applied, such as when spreading the formulation onto the oral mucosa, the structure begins to flow. Once the force is removed, the viscosity increases again due to new entanglements and interactions through the Brownian motion of the macromolecules [43], preventing the formulation from flowing away from the oral mucosa [42].

Moreover, the viscosity recovery of FL100 and FLE100 occurred more slowly (greater thixotropy) than F100, indicating that FL100 and FLE100 have a more compact and stable structure that requires more energy to reorganize [42]. This high level of structural organization may be related to ionic interactions between the cationic charges of lidocaine and the negative charge of saliva, resulting in increased cross-linking [43]. The increase in cross-linking also reflected in the responses found in oscillatory rheology, with formulations presenting elastic properties, high gel strength, and low loss tangent upon dilution with saliva. Therefore, FL and FLE appear to have adequate properties when subjected to stress similar to in vivo application, providing important information regarding their suitability for clinical use [44], especially in the dental area.

The commercial formulation presented opposite rheological characteristics. It is an ointment based on macrogol, a polymer commonly used as a viscosity modifier in lipophilic vehicles. When diluted in an aqueous solvent such as saliva, hydrogen bonds form between water molecules and the polymer’s hydroxyl groups, increasing its solubility and consequently reducing viscosity [45]. This property may diminish its mucoadhesion in a humid environment like the oral cavity, as confirmed in the in vitro mucoadhesion assay (Table 2).

Texture profile analyses of diluted commercial formulation (FC100) and microemulsions (F, FL and FLE) were not performed, as these formulations are fluids with high loss tangent values, making it difficult to measure hardness, compressibility, adhesiveness, and cohesion [46]. After incorporating 100% saliva, the microemulsions (F, FL and FLE) became viscous formulations with increased hardness, compressibility, and adhesiveness, which also increased with the concentration of saliva [20]. However, the mechanical properties of these formulations after contact with 100% saliva (F100, FL100 and FLE100) were lower than those of the FC without saliva. Considering the humid environment provided by saliva [47] and the loss of viscosity of the FC upon contact with 100% saliva, this mechanical behavior is not expected to interfere with the clinical application of the developed formulations, as indicated by the findings (Table 2). Nevertheless, the in vivo efficacy of these formulations remains to be evaluated in future studies, which can corroborate such in vitro findings.

As indicated by the findings on phase behavior, rheology, and texture profile, the mucoadhesion assay aligns with these results, demonstrating an enhanced mucoadhesive work for FL and FLE in the presence of 30% or 100% of saliva (FL30, FL100, FLE30, and FLE100). In contrast, diluting the commercial formulation with 100% saliva resulted in a threefold reduction in its mucoadhesion, a behavior not observed with the developed formulations.

The work of mucoadhesive force is the most appropriate parameter for comparing mucoadhesion between different formulations, as it reflects the sum of forces acting on the formulation in situ and measures the energy required to detach the two surfaces [48]. The mucoadhesion mechanism may involve a swelling process, where the formulation absorbs water from the mucosa, swells, and forms a strong mucoadhesive bond, as well as an electronic mechanism, driven by the electrostatic attraction between chitosan and saliva [49]. The more pronounced increase in mucoadhesion observed in FLE100 may be attributed to the rheological characteristics of FL100, as high viscosity and gel strength can limit polymer flexibility and mobility, thereby reducing diffusion and entanglement with the mucosa [50]. Nevertheless, the enhanced mucoadhesion of FL100 and FLE100 could potentially prolong the residence time at the site of action and improve therapeutic efficacy while reducing the required dose and frequency of administration [51], particularly for pain management. However, while these in vitro findings suggest such benefits, further in vivo studies are necessary to confirm their clinical relevance.

The release profile of FC and FL showed that both formulations controlled lidocaine release, but the commercial formulation released five times more lidocaine than FL over 24 h. The developed microemulsion promoted sustained release, as demonstrated for other nanostructured delivery systems [52] and liquid crystals with other active ingredients [23]. The marked reduction in lidocaine release and permeation observed for FL (up to 93% lower permeation flow compared to FC) reflects its function as a controlled drug delivery system. This behavior may contribute to a prolonged anesthetic effect, reducing the need for repeated applications, preventing high peak concentrations, lowering the risk of adverse effects, and maintaining more stable anesthetic levels over time [53,54].

Importantly, despite the lower overall permeation, FL exhibited immediate onset of permeation, likely due to the presence of excipients with known permeation-enhancing properties [25]. This characteristic may result in a faster onset of topical anesthesia [25,43], while simultaneously increasing drug retention at the application site. Such a combination, early onset with sustained effect is clinically valuable in the management of painful oral conditions, such as oral mucositis, and other situations where topical anesthetics can enhance patient comfort during everyday activities like swallowing, chewing, and speaking. 

Even so, given the in vitro nature of this study, validation in in vivo models is essential to confirm whether the observed permeation dynamics translate into clinical parameters of onset, depth, and duration of anesthesia. Studies in animal models or human volunteers could assess key outcomes such as time to onset, duration of analgesia, depth of tissue penetration, and systemic exposure. Previous studies have demonstrated that in vitro permeation flux can be predictive of in vivo systemic absorption and pharmacokinetic behavior of topical lidocaine formulations. For example, a study demonstrated that increasing the drug load in a nonaqueous topical system led to proportional increases in both in vitro permeation and plasma lidocaine levels in human volunteers [55]. Similar findings have been reported with other nanostructured drug delivery systems, where modifications to carrier architecture resulted in enhanced drug retention and controlled release, ultimately improving therapeutic performance in vivo [56]. These findings support the relevance of permeation parameters—such as flux and lag time—as indicators of clinical performance, and reinforce the potential of the FL formulation to achieve sustained anesthetic effect with reduced systemic exposure.

Investigating the toxicity of lidocaine incorporated into a new nanostructured drug delivery system is necessary, as this new system can alter the toxicological profile of lidocaine compared to commercial formulations. In this context, the chorioallantoic membrane (CAM) assay has emerged as a valuable alternative in vivo model, as it avoids the use of sentient animals and has been increasingly reported for assessing the toxicity of drug delivery systems [57] and topical anesthetics [29]. Its application allows for an initial screening of mucosal irritancy and vascular responses, contributing to the refinement of preclinical evaluation strategies [58]

In our study we demonstrated that the developed formulations presented similar toxicity than the commercial lidocaine formulation (moderate irritant). This classification is based on the evaluation of three vascular endpoints observed on the chorioallantoic membrane: hemorrhage (bleeding), lysis (cell membrane rupture), and coagulation (clot formation). Each of these reactions is scored on a scale from 0 to 3, where 0 indicates no reaction and 3 indicates a strong reaction. The final irritancy score is obtained by summing the individual scores for these three endpoints. According to the ECVAM protocol, a total score between 6 and 12 corresponds to a moderate irritant. The similarity in scores between the test and commercial formulations suggests that the incorporation of lidocaine and epinephrine into the liquid crystal precursor system did not increase the irritant potential compared to the reference product [27].

Other commercially available dental topical anesthetics, such as those based on lidocaine, benzocaine, and EMLA (a eutectic mixture of lidocaine and prilocaine), have also been classified as moderately irritating [59]. These findings suggest that liquid crystal precursor systems show promise as non-toxic and safe candidates for future in vivo studies.

The HET-CAM model evaluates acute vascular responses, such as hemorrhage, lysis, and coagulation, which are particularly relevant for formulations intended for mucosal application. While it offers a reliable and ethically accepted alternative to traditional animal testing, it does not fully replicate the complexity of mammalian tissues, especially regarding immune responses, metabolic processing, and long-term exposure. Therefore, although the assay provides valuable preliminary safety data, further in vivo studies in mammalian models are necessary to confirm the biocompatibility and clinical safety of the formulations.

It is important to emphasize that the lack of in vivo efficacy data is a recognized limitation of the present study, and despite our main objective being to provide proof of concept and detailed characterization of an innovative topical anesthetic, in vivo studies are necessary to confirm its anesthetic efficacy. In this sense, future investigations should include well-designed in vivo studies using appropriate animal models to evaluate anesthetic effectiveness, local tolerability, and systemic absorption, in order to better replicate the complexity of mammalian tissues. These studies will be fundamental to validate the pharmacodynamic effects and biocompatibility of the formulation. Subsequently, clinical trials will be required to assess safety, efficacy, and patient acceptability in dental procedures. Further studies on long-term stability and scalability will also be essential to support regulatory approval and eventual clinical application.

Finally, although this study focuses on lidocaine, the formulation’s principles can be applied to other drugs with similar delivery requirements such as corticosteroids, anti-inflammatory agents, and antibiotics for treating various oral conditions. The liquid crystalline system used here provides a versatile platform that can be customized to enhance the bioavailability and controlled release of various active pharmaceutical ingredients. Future research should explore its adaptation for other drugs, potentially expanding its applications in buccal drug delivery.

## 4. Materials and Methods

### 4.1. Materials

PPG-5-CETETH-20 (Procetyl^®^ AWS) was donated by Croda (Snaith, Goole, UK). Lidocaine hydrochloride (LH) was donated by Cristalia (Itapira, Sao Paulo, Brazil). Oleic acid was purchased from Synth (Diadema, Sao Paulo, Brazil). Epinephrine (E) and low-molecular-weight chitosan were purchased from Sigma Aldrich (Steinheim, North Rhine-Westphalia, Germany). The commercial formulation (FC or FCom) is lidocaine 50 mg/g dermatological ointment orange flavor from EMS Pharma (Hortolandia, Sao Paulo, Brazil).

### 4.2. Preparation of Liquid Crystal Precursor System

The development of the liquid crystalline precursor system (LCPS) was initiated by constructing a phase diagram using PPG-5-CETETH-20 and oleic acid as the oil phase, and a 0.5% (*w*/*v*) chitosan dispersion as the aqueous phase. The oil phase was prepared by manually mixing PPG-5-CETETH-20 with oleic acid. This mixture was then slowly added to the aqueous phase under magnetic stirring at 150 rpm for 10 min. The formulation without drugs was referred to as the blank LCPS, or F.

Formulations loaded with LH (FL) and co-loaded with LH and epinephrine (FLE) were prepared following the same procedure as the base formulation (F), with the active compounds incorporated into the process. Specifically, 5% (*w*/*w*) LH was introduced into the aqueous phase, while 0.001% (*w*/*w*) epinephrine was added to the oil phase. The two phases were then combined using the method previously described for F. The concentration of LH (5%) was selected based on typical concentrations found in commercial topical formulations.

All formulations were allowed to rest for 48 h to stabilize and eliminate bubbles.

Artificial saliva was added to the formulations in two concentrations: 30% and 100% (*v*/*v*), resulting in dilution ratios of 3:10 and 1:1 (artificial saliva to formulation), respectively. These were designated as F30, FL30, FLE30 and F100, FL100, FLE100. In a pilot study, several concentrations were tested, and phase transition was observed at both 30% and 100% dilution, which were selected for further characterization. The chosen concentrations (30% and 100% dilution with artificial saliva), similarly employed in previous studies from our group, were observed to induce phase transition, which is critical for the formulation’s performance. This phase transition ensures that the formulation maintains its structure and effectiveness in the oral environment, leading to optimal drug delivery [20,22].

The artificial saliva (pH 6.8) was composed of 8 g/L of sodium chloride (NaCl), 0.19 g/L of potassium monobasic phosphate (KH_2_PO_4_), and 2.28 g/L of disodium phosphate (Na_2_HPO_4_), as previously described [60]. After saliva addition, the systems were classified as the Transparent Liquid System (TLS), Transparent Viscous System (TVS), Opaque System (OS), and Phase Separation (PS) for plotting the phase diagram.

All LCPS formulations (F, FL, FLE, F100, FL100, and FLE100) were freshly prepared immediately prior to use, following the described protocol, and handled under light-protected conditions to minimize potential degradation of epinephrine. No antioxidants or stabilizers were incorporated into the formulations, as the experimental design did not involve long-term storage. The commercial formulation (FC) used was lidocaine 50 mg/g orange flavor dermatological ointment from EMS.

### 4.3. Structural Characterization of Formulations

#### 4.3.1. Polarized Light Microscopy (PLM)

A small aliquot of each formulation (F, FL, FLE, F30, FL30, FLE30, F100, FL100, and FLE100) was placed on a glass slide, which was covered with a coverslip and analyzed under a polarized light microscope (Olympus BX41, Olympus America Inc., Center Valley, PA, USA) at 20× magnification.

#### 4.3.2. Small-Angle X-Ray Scattering (SAXS)

SAXS studies were conducted using the Xenocs XEUSS™ equipment (Xenocs, Sassenage, France).. Radiation was generated by a GENIX™ source (Cu Kα edge, λ = 1.54 Å) with the beam focused by a FOX2D™ optic. Beam collimation was achieved using two sets of scatter-free 0.49 mm^2^ slits (Xenocs 2.0) in high-resolution mode, and data was collected by a Dectris Pilatus™ 300k detector (Dectris, Baden, Switzerland). The sample-to-detector distance was 0.97 mm, providing a q range of 0.15 < q < 4.5 nm^−1^, where q is the reciprocal spatial momentum transfer modulus defined as (q = (4π sinθ)/λ, with 2θ being the scattering angle and λ the wavelength of the radiation.

All formulations were stored in borosilicate glass capillaries. Measurements for F, FL, FLE, F30, FL30, and FLE30 were conducted in eight-frame 900 capillaries, while formulations F100, FL100, and FLE100 were measured in mica for 300 s (4 × 300 s, 7 × 300 s, and 12 × 300 s).

#### 4.3.3. Rheological Analysis

The rheological analysis of the formulations was performed using a Discovery Hybrid Rheometer DHR-1 rheometer (TA Instruments Ltd., Leatherhead, UK) with a 40 mm plate diameter and a 500 μm gap. Samples were carefully placed on the lower plate of the rheometer and allowed to rest for 30 s before analysis.


**Continuous rheological analysis**


The shear rate ranged from 0.01 to 100 s^−1^ for the ascending curve and from 100 to 0.01 s^−1^ for the descending curve, each for 120 s at 37 °C. Data from the ascending curves was fitted to the Herschel–Bulkley rheological model, with continuous rheological parameters obtained using Equation (1):*τ* = *k* × *y*^η^
(1)
where τ is shear stress (Pa); k is consistency coefficient (Pa.sn); γ is shear rate (s-1); and η is fluid behavior index (dimensionless).


**Oscillatory rheological analysis**


The stress sweep test determined the viscoelastic region, using a shear stress range of 0.1 to 100 Pa and a frequency of 1 Hz. After determining the tension in the viscoelastic region for each formulation, the frequency sweep test was performed to determine the elastic modulus (G′) and viscous modulus (G″). The frequency range was 0.1 to 10 Hz at the previously determined viscoelastic region tension, at 37 °C. Relationships between the modules and oscillatory frequency were used to determine G′ and loss tangent (tan δ) with HR1—5332-1380: TA Instruments Trios v.4.1.0.31739 software. Structural strength and viscoelastic exponent were calculated using Equation (2):G′ = S × ω^n^
(2)
where G′ is the storage modulus, ω is the oscillatory frequency and S is the gel resistance, with n being the viscoelastic exponent. The analysis was performed with three independent samples.

### 4.4. Texture Profile Analysis

The texture profile of formulations F30, FL30, FLE30, F100, FL100, and FLE100 was analyzed using a TA-XT plus texture analyzer (Stable Micro Systems, Godalming, UK). For analysis, 10 g of each formulation was placed in 50 mL conical centrifuge tubes and centrifuged at 4000 rpm for 3 min to eliminate air bubbles and smooth the surface. The tubes were positioned under the analytical probe (10 mm diameter) of the texture analyzer, programmed to compress the sample at 2 mm/s to a predefined depth (15 mm) before returning to the surface at the same speed. After a 15 s rest period, a second compression was performed under identical conditions [61,62]. Parameters such as hardness (mN), compressibility (mN.s), adhesiveness (mN.s), and cohesion (dimensionless) were calculated from the force-time curves [62]. All analyses were conducted in ten replicates at room temperature (25.0 ± 0.5 °C).

### 4.5. In Vitro Evaluation of the Mucoadhesive Force

The mucoadhesive force of the formulations was analyzed using porcine buccal mucosa epithelium. Porcine jaws (*Landrace domestic Susscrofa*, five months old, weighing 75–80 kg) were obtained from a local slaughterhouse (Angelelli^®^ Ltda., Piracicaba—SP, Brazil, certified by the Secretariat of Agriculture and Supply of the State of São Paulo—SIF 2259). The buccal mucosa was separated from adjacent tissue with a scalpel and immersed in phosphate-buffered saline (pH 7.4) at 60 °C for 2 min. The epithelium was then carefully separated from the connective tissue using a Molt detacher and visually inspected for tissue injuries to exclude damaged samples [9]. The epithelium was attached to the mucoadhesion device and kept in artificial saliva (pH 6.8, 37 °C) before the permeation studies. The mucoadhesion test was performed using a TA-XT Plus Texture Analyzer (Stable Micro Systems, UK) equipped with a 5 kg load cell. The epithelium was fixed to the lower probe of a custom-designed mucoadhesion device (A/MUC), while the formulation was placed on the upper probe. The device includes a central groove to standardize the volume of formulation applied. The test was conducted at 37 °C, with a pre-test speed of 2 mm/s, a contact force of 0.552 N, and a contact time of 10 s. Detachment was performed at a post-test speed of 2 mm/s. Mucoadhesive parameters, including peak mucoadhesive force (mN) and mucoadhesion work (mN·s), were calculated from the force–time curve using Exponent^®^ software (Exponent For The XTPlus Version 6, 1, 18, 0). Each formulation was tested in ten independent replicates. Illustrative figures of the experimental setup and device are provided in the Appendix A.

### 4.6. In Vitro Release Study

The in vitro release study of lidocaine from FL and FC was conducted using Manual Transdermal equipment (Hanson Research Corporation; Chatsworth, CA, USA) with Franz diffusion cells (7 mL volume, 1.77 cm^2^ permeation area). A synthetic cellulose acetate membrane (12–14 kDa MWCO) and a receiving solution of phosphate-buffered saline (PBS, pH 7.4, 37 °C, 300 rpm stirring) were used. Formulations were transferred to the dosing ring in the donor compartment of the Franz diffusion cell under infinite dose conditions. Samples were collected from the receiving compartment at predetermined times (15 min, 30 min, 45 min, 1 h, 1.5 h, 2 h, 2.5 h, 3 h, 4 h, 5 h, 6 h, 7 h, 12 h, 24 h).

Lidocaine hydrochloride in F and released from F samples was quantified by HPLC using a previously validated method [63]. The mobile phase was a 60:40 (*v*/*v*) mixture of acetonitrile and 25 mM NH_4_OH, adjusted to pH 7.0 with H_3_PO_4_, at a flow rate of 1.2 mL/min. A reversed-phase column (150 × 4.60 mm, 5 μm, Phenomenex, Torrance, CA, USA) was used, with a 20 μL injection volume and a 220 nm detector wavelength.

Lidocaine release profiles were calculated using Equation (3):Q = Ct × Vr + Σ Vc × Cc (3)
where Q (μg.cm^2^) represents the total amount of lidocaine hydrochloride permeated up to time t; Ct (µg.mL^−1^ × cm^2^) is the lidocaine hydrochloride concentration measured at time t; Vr (mL) is the receptor solution volume (7 mL); Cc (μg.mL^−1^ × cm^2^) is the concentration at the previous sampling, and Vc (mL) is the volume sampled [20].

Lidocaine hydrochloride release profiles were fitted to various kinetic models (zero order, first order, Korsmeyer-Peppas, Higuchi) to understand the release mechanisms.

### 4.7. In Vitro Permeation Study

The permeation study was conducted using Manual Transdermal equipment (Hanson Research Corporation; Chatsworth, CA, USA) with Franz diffusion cells and fresh buccal epithelium, prepared as previously described. Formulations FL (0.5 g) and FC (0.7 g) were placed in the dosing ring located in the donor compartment of the Franz diffusion cell under infinite dose conditions. Samples (0.4 mL) were collected from the receiving compartment at predetermined intervals (15, 30, 45, 60, 90, 120, 150, 180 min). The permeated lidocaine hydrochloride (LH) was quantified by HPLC. After quantification, individual graphs were generated for each vertical diffusion cell by plotting the amount of lidocaine accumulated in the receptor compartment against time. The steady-state flux (*J_ss_*) was calculated according to Fick’s first law, as described in Equation (4).(4)$$J_{ss}=\frac{{\Delta Q}_{t}}{\Delta_{t}\times A}\left[\mu g\cdot {\mathrm{cm}}^{-2}\times \min^{-1}\right]$$
where ΔQt is the difference in the amount of drug permeated, Δt is the difference in the measurement time points (min), and A is the permeation area (cm^2^).

The lag time was determined from the *x*-axis intercept of this straight line, indicating the time required for drug permeation to commence, signifying a positive drug concentration in the receptor chamber thereafter. This mathematical model is widely recognized for its application in permeation assays and comparative studies [35].

### 4.8. In Vivo Chicken Chorioallantoic Membrane (CAM) Toxicity Assay

The Hen’s Egg Test on the Chorioallantoic Membrane (HET-CAM) was performed following Protocol No. 96, as published by the European Centre for the Validation of Alternative Methods [27,28]. This method is widely recognized for assessing the irritation potential of chemical formulations as an alternative to traditional in vivo models.

According to this protocol, fertilized chicken eggs on the 3rd day of development were visually inspected to assess shell integrity and weight. Eggs weighing 50–60 g were selected, cleaned with 70% ethyl alcohol, and incubated at 37.5 °C and 55–65% relative humidity (Galinha Choca Chocadeiras & Acessórios, São Bernardo do Campo—SP, Brazil). A window was opened in the egg air chamber to evaluate the CAM on the tenth day of incubation. A photographic record of the CAM was taken before treatments (time T0). Subsequently, 0.3 mL of each sample—(i) positive control solution (0.1 mol/L sodium hydroxide, NaOH), (ii) negative control solution (0.9% saline solution), or (iii) test formulations (F, FL, FLE, and FC)—was applied directly to the CAM for 180 s (T5). After the application period, the samples were carefully rinsed with 20 mL of saline solution. The CAM was then evaluated and photographed 3 min after rinsing to assess the final effect of the treatments.

Formulations were classified as non-irritating, slightly irritating, moderately irritating, irritating, or severely irritating based on the sum of the treatment scores for each formulation [28], as illustrated in Table 5. Scores were obtained from possible reactions (hemorrhage, lysis, and clot), individually classified into three categories: 0 (no reaction), 1 (weak reaction), 2 (moderate reaction), and 3 (strong reaction). In addition, representative images for each irritancy category are included in Table 5 to visually illustrate the classification system applied in this study, based on typical endpoints observed in CAM tissue.

Although in vitro cytotoxicity assays using oral epithelial or gingival cell lines (e.g., MTT or LDH assays) are commonly employed to assess biocompatibility, we opted for the HET-CAM assay due to its recognized validity as an alternative in vivo model. The HET-CAM test is endorsed by the European Centre for the Validation of Alternative Methods (ECVAM) and is widely used to evaluate the irritation potential of formulations applied to mucosal tissues. This model offers a reliable, ethically accepted, and animal-free approach to assess acute vascular responses such as hemorrhage, lysis, and coagulation, which are highly relevant for formulations intended for oral mucosal application. Therefore, the use of the HET-CAM assay aligns with current regulatory and ethical standards for preclinical safety evaluation.

### 4.9. Statistical Analysis

The rheological study data met the assumptions of normality and homoscedasticity, as evaluated by the Shapiro–Wilk and Levene tests, respectively. Consequently, the data was analyzed using descriptive statistics followed by two-way ANOVA, considering ‘Formulations’ at four levels and ‘Saliva’ at three levels, using the Bonferroni post hoc test for multiple comparisons.

The texture profile analysis (TPA) data met the normality assumption evaluated by the Shapiro–Wilk test across all groups for the four variables (*p* = 0.062), except for one group in the Hardness variable (*p* = 0.037) and one group in the Adhesiveness variable (*p* = 0.039). All data exhibited heteroscedasticity according to the Levene test (*p* < 0.003). Therefore, the data was subjected to one-way ANOVA with Welch’s correction, followed by the Games–Howell post hoc test. Confidence intervals for the means were also estimated.

The peak of the mucoadhesive force (N) and mucoadhesion work (N × s) were analyzed using descriptive statistics, followed by two-way ANOVA, considering ‘Formulations’ at four levels and ‘Saliva’ at three levels. The normality assumption was met in 12 groups for the peak mucoadhesive force variable (*p* = 0.063) and in 9 groups for the mucoadhesion work variable (*p* = 0.051), according to the Shapiro–Wilk test. The data exhibited heteroscedasticity according to the Levene test (*p* < 0.001). Therefore, when ANOVA detected statistically significant effects, multiple comparisons of means were performed, pairwise, using the Games–Howell post hoc test.

Permeation parameters data was compared using the unpaired *t*-test.

A significant level of 0.05 was adopted for all analyses, which were conducted using IBM^®^ SPSS^®^ Statistics software (version 26).

## 5. Conclusions

We successfully developed a mucoadhesive liquid crystal precursor system for the delivery of lidocaine and epinephrine, demonstrating promising attributes for topical anesthesia in the oral cavity. Structural analysis highlighted its dynamic behavior, transitioning into liquid crystalline structures upon dilution with saliva. Rheological studies confirmed thixotropic behavior, indicating adaptability to shear stress, along with enhanced mucoadhesive properties in the presence of saliva. In vitro release and permeation studies suggested a sustained drug release profile. Importantly, the developed formulations demonstrated a safety profile comparable to the commercial formulation in the CAM model. However, as this study was designed as a proof of concept, further comprehensive in vivo efficacy studies are necessary to confirm its clinical benefits.

## Figures and Tables

**Figure 1 pharmaceuticals-18-01166-f001:**
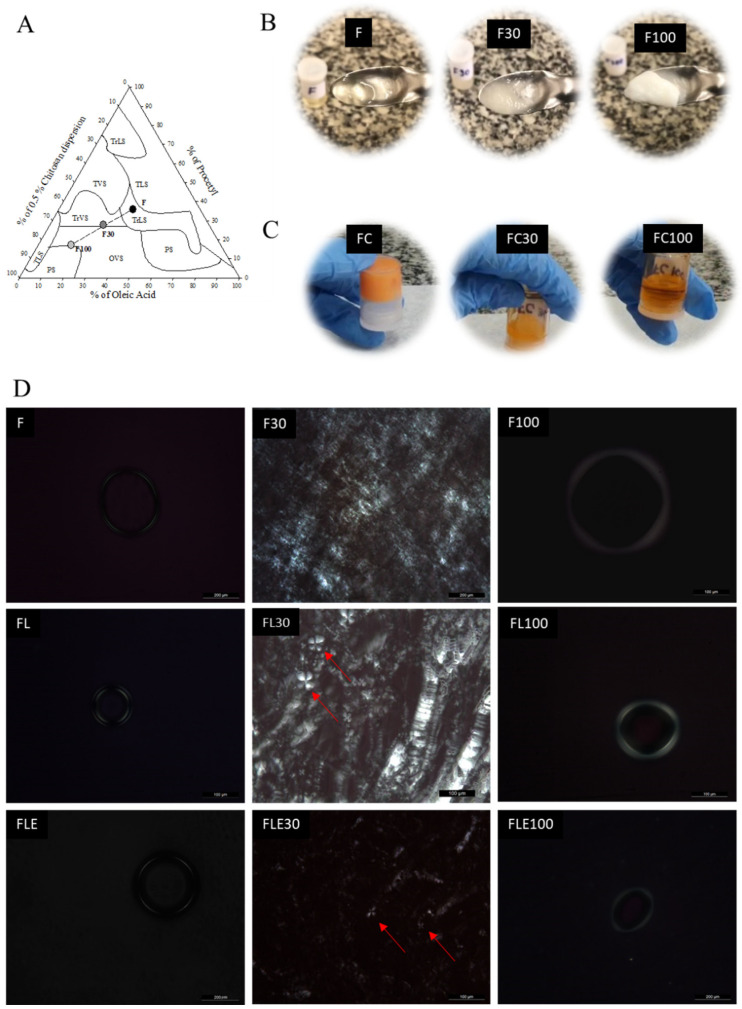
Development and structural characterization of formulations. (**A**) Ternary diagrams composed of Procetyl^®^ AWS, oleic acid, and 0.5% (*w*/*v*) chitosan dispersion, with Transparent Liquid System (TLS), Translucent Liquid System (TrLT), Transparent Viscous System (TVS), Translucent Viscous System (TrVT), Opaque Viscous System (OVS), and Phase Separation (PS). F is the precursor of the liquid crystalline system; F30 is the formulation F diluted with 30% artificial saliva; and F100 is the formulation F diluted with 100% artificial saliva. (**B**) Visual aspects of F, FC, F30, FC30, F100, and FC100. (**C**) Visual aspects of the commercial formulation (FC), FC diluted with 30% artificial saliva (FC30), and FC diluted with 100% of artificial saliva (FC100). (**D**) Photomicrographs of formulations F, F30, F100, FL, FL30, FL100, FLE, FLE30, and FLE100. Magnification at 20× (F, F30, FLE, and FLE100) or 40× (F100, FL, FL30, FL100, FLE30). FL is the liquid crystal precursor system with 5% lidocaine hydrochloride. FLE is the liquid crystal precursor system (LPCS) with 5% lidocaine hydrochloride and 0.001% epinephrine. FC is the commercial formulation of lidocaine 50 mg/g EMS orange flavor dermatological ointment. The formulations were diluted with 30% (F30, FL30, FLE30, and FC30) and 100% of artificial saliva (F100, FL100, FLE100, and FC100). The red arrows point to the Maltese cross formation.

**Figure 2 pharmaceuticals-18-01166-f002:**
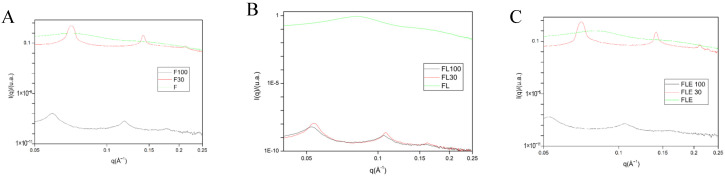
Structural evaluation of formulations by SAXS: (**A**) F, F30, and F100; (**B**) FL, FL30, and FL100; and (**C**) FLE, FLE30, and FLE100. F is the liquid crystal precursor system without the incorporation of lidocaine and epinephrine. FL is the liquid crystal precursor system with 5% lidocaine hydrochloride. FLE is the liquid crystal precursor system with 5% lidocaine hydrochloride and 0.001% epinephrine. The formulations were diluted with 30% (F30, FL30, and FLE30) and 100% (F100, FL100, and FLE100) of artificial saliva.

**Figure 3 pharmaceuticals-18-01166-f003:**
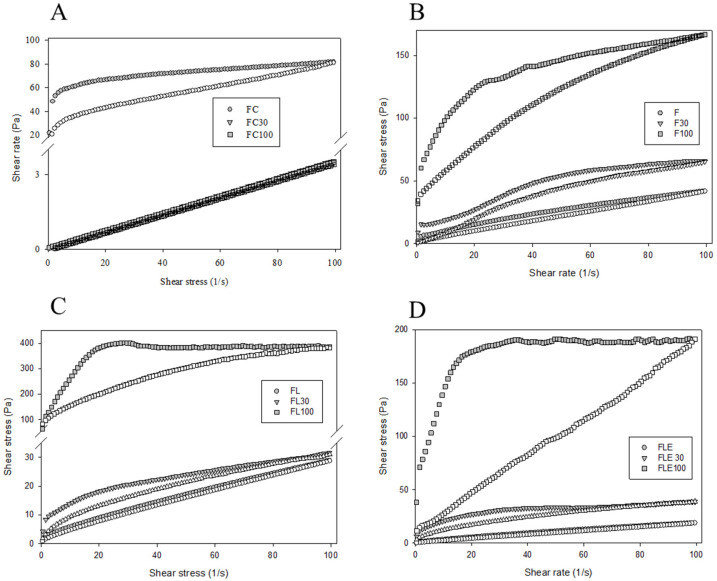
Flow rheograms of the different formulations evaluated. (**A**) FC, (**B**) F, (**C**) FL, and (**D**) FLE. FC is the commercial formulation of lidocaine 50 mg/g EMS orange flavor dermatological ointment. F is the liquid crystal precursor system without the incorporation of lidocaine hydrochloride and epinephrine. FL is the liquid crystal precursor system with 5% lidocaine hydrochloride. FLE is the liquid crystal precursor system with 5% lidocaine hydrochloride and 0.001% epinephrine. The formulations were diluted with 30% (F30, FL30, FLE30, and FC30) and 100% (F100, FL100, FLE100, and FC100) of artificial saliva. Filled symbols are ascending curve and empty symbols are descending curve. Standard deviations have been omitted for clarity; however, in all cases, the coefficient of variation in the triplicate analyses was less than 10%. Data was collected at 37 ± 0.5 °C.

**Figure 4 pharmaceuticals-18-01166-f004:**
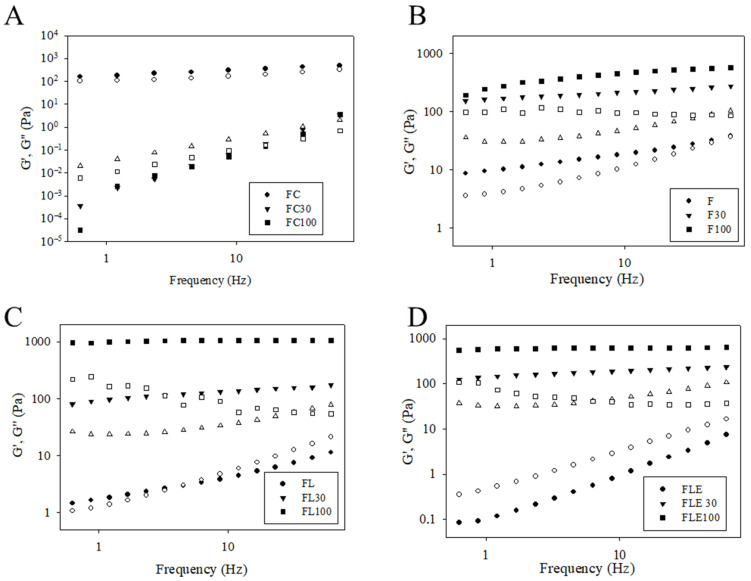
Variation in the storage modulus G’ (full symbols) and loss G’’ (empty symbols) as a function of frequency for the different formulations evaluated. (**A**) FC, (**B**) F, (**C**) FL, and (**D**) FLE. FC is the commercial formulation of lidocaine 50 mg/g EMS orange flavor dermatological ointment. F is the liquid crystal precursor system without the incorporation of lidocaine hydrochloride and epinephrine. FL is the liquid crystal precursor system with 5% lidocaine hydrochloride. FLE is the liquid crystal precursor system with 5% lidocaine hydrochloride and 0.001% epinephrine. The formulations were diluted with 30% (F30, FL30, FLE30, and FLE100) and 100% (F100, FL100, FLE100, and FC100) of artificial saliva. Standard deviations have been omitted for clarity; however, in all cases, the coefficient of variation in the triplicate analyses was less than 10%. Data was collected at 37 ± 0.5 °C.

**Figure 5 pharmaceuticals-18-01166-f005:**
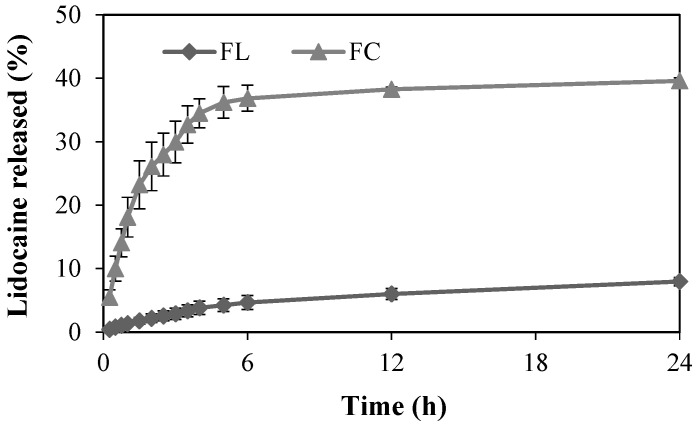
In vitro release profile of lidocaine from FC and FL in 24 h at 37 °C, *n* = 6. FL is the liquid crystal precursor system with 5% lidocaine hydrochloride. FC is the commercial formulation of lidocaine 50 mg/g orange flavor dermatological ointment from EMS company.

**Figure 6 pharmaceuticals-18-01166-f006:**
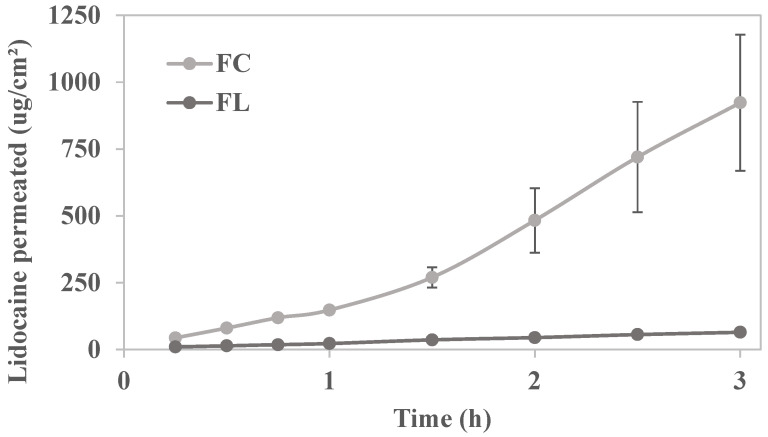
In vitro permeation profile of lidocaine hydrochloride across porcine buccal mucosa incorporated in FL and FC for 3 h (*n* = 3–6). FL is the liquid crystal precursor system with 5% lidocaine hydrochloride. FC is the commercial formulation of lidocaine 50 mg/g orange flavor dermatological ointment from EMS company. Linear regression analysis between curves (*p* < 0.0001).

**Figure 7 pharmaceuticals-18-01166-f007:**
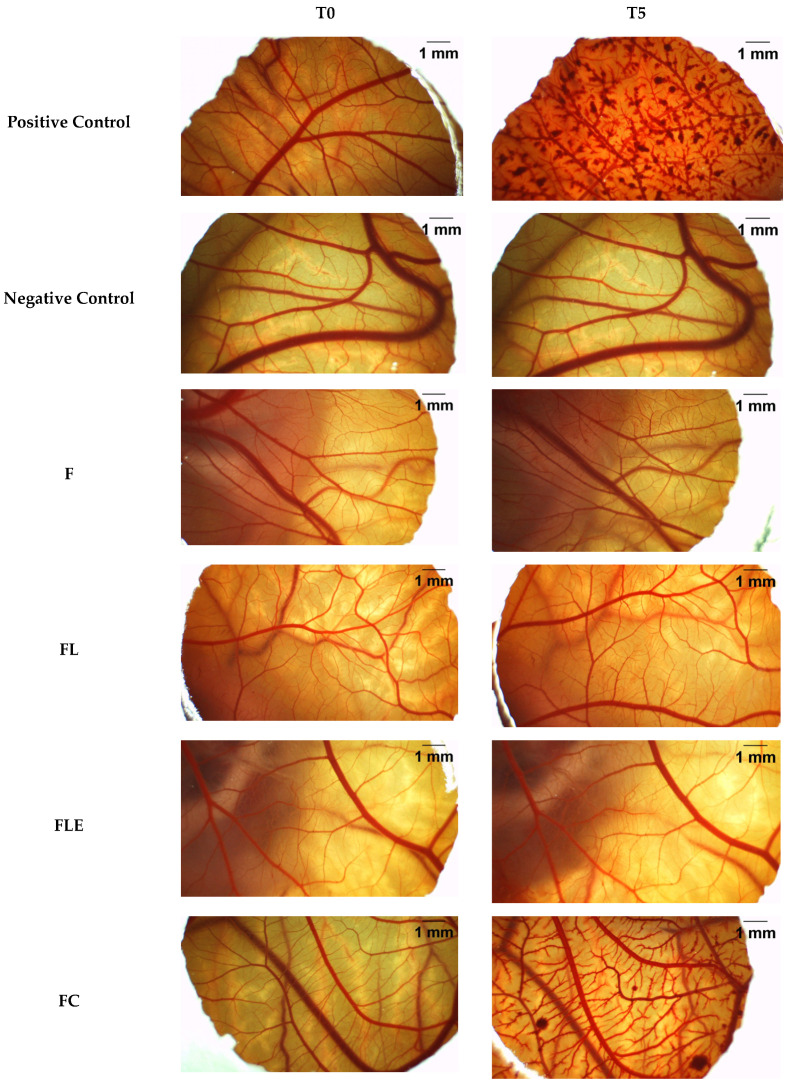
Representative images of the CAM before (T0) and after 5 min (T5) of application of the test substances (positive control, negative control, F, FL, FLE, and FC to evaluate possible toxic effects. F is the liquid crystal precursor system without the incorporation of lidocaine hydrochloride and epinephrine. FL is the liquid crystal precursor system with 5% lidocaine hydrochloride. FLE is the liquid crystal precursor system with 5% lidocaine hydrochloride and 0.001% epinephrine. FC is the commercial formulation of lidocaine 50 mg/g EMS orange flavor dermatological ointment.

**Table 1 pharmaceuticals-18-01166-t001:** Mechanical properties (hardness, compressibility, adhesion, and cohesion) and mucoadhesive properties (peak of mucoadhesive strength and mucoadhesive strength work) of formulations.

Formulations	Mechanical Properties				Mucoadhesive Properties	
	Hardness (mN)	Compressibility (mN·s)	Adhesiveness (mN·s)	Cohesion (Dimensionless)	Peak of Mucoadhesive Strength (mN)	Mucoadhesive Strength Work (mN·s)
F	- *	- *	- *	- *	680 ± 73 ^g^	189 ± 28 ^d^
F30	49 ± 2 ^b^	147 ± 7 ^b^	81 ± 10 ^b^	0.687 ± 49 ^a^	466 ± 123 ^ef^	223 ± 29 ^d^
F100	138 ± 9 ^c^	478 ± 40 ^d^	464 ± 43 ^e^	0.803 ± 17 ^b^	144 ± 23 ^a^	174 ± 18 ^cd^
FL	- *	- *	- *	- *	238 ± 24 ^d^	84 ± 13 ^a^
FL30	33 ± 1 ^a^	111 ± 5 ^a^	30 ± 5 ^a^	0.821 ± 21 ^b^	183 ± 22 ^bc^	119 ± 22 ^bc^
FL100	179 ± 11 ^d^	681 ± 52 ^e^	596 ± 44 ^e^	0.742 ± 15 ^a^	218 ± 24 ^cd^	111 ± 7 ^bc^
FLE	- *	- *	- *	- *	420 ± 51 ^e^	120 ± 18 ^bc^
FLE30	34 ± 1 ^a^	98 ± 8 ^a^	54 ± 5 ^b^	0.793 ± 26 ^b^	407 ± 178 ^e^	210 ± 24 ^d^
FLE100	191 ± 24 ^d^	756 ± 111 ^e^	638 ± 95 ^e^	0.711 ± 26 ^a^	154 ± 18 ^ab^	296 ± 21 ^e^
FC	357 ± 32 ^e^	1005 ± 131 ^f^	1276 ± 262 ^f^	0.933 ± 52 ^c^	992 ± 284 ^h^	313 ± 79 ^e^
FC30	- *	- *	- *	- *	600 ± 87 ^f^	83 ± 13 ^a^
FC100	- *	- *	- *	- *	640 ± 140 ^fg^	99 ± 17 ^ab^

Different lowercase letters denote averages with statistically significant differences, according to the Games–Howell post-test (*p* < 0.05). Each value represents the mean ± standard deviation of 10 replicates. * Liquid samples without hardness, compressibility, adhesion, and cohesion values. F is the liquid crystal precursor system without the incorporation of lidocaine and epinephrine. FL is the liquid crystal precursor system with 5% lidocaine. FLE is the liquid crystal precursor system with 5% lidocaine and 0.001% epinephrine. FC is the commercial formulation of lidocaine 50 mg/g EMS orange flavor dermatological ointment. The formulations were diluted with 30% (F30, FL30, FLE30, and FC30) and 100% of artificial saliva (F100, FL100, FLE100, and FC100).

**Table 2 pharmaceuticals-18-01166-t002:** Adjusted parameters of the kinetic models used in the release of lidocaine hydrochloride.

Kinetic Models	Formulations	
	FC	FL
Korsmeyer-Peppas		
R^2^	0.8209	0.9704
n	0.2765	0.5105
Higuchi		
R^2^	0.3968	0.9723
K	1.5917	1.6935
First order		
R^2^	0.000	0.4965
K	0.0009	0.0046
Weibull		
R^2^	0.9979	0.9943
b	0.7457	0.7780

FL is the liquid crystal precursor system with 5% lidocaine hydrochloride. FC is the commercial formulation of lidocaine 50 mg/g orange flavor dermatological ointment from EMS company.

**Table 3 pharmaceuticals-18-01166-t003:** In vitro permeation parameters calculated from individual lidocaine permeation profiles.

Parameters	Formulations	
	FC	FL
Flow (µg.cm^−2^·h^−1^)	635.0 ± 161.4 *	41.4 ± 1.01
Lag time (h)	0.33 ± 0.03 *	-

* *p* < 0.05, *t*-test. Data represents mean ± SD (n = 6). Parameters were analyzed separately. FL is the liquid crystal precursor system with 5% lidocaine hydrochloride. FC is the commercial formulation of lidocaine 50 mg/g orange flavor dermatological ointment from EMS company.

**Table 4 pharmaceuticals-18-01166-t004:** Toxicity classification of F, FL, FLE, FC, positive control (sodium hydroxide (NaOH at 0.1 mol/L), and negative control (0.9% saline solution) based on CAM assay. F is the liquid crystal precursor system without the incorporation of lidocaine hydrochloride and epinephrine. FL is the liquid crystal precursor system with 5% lidocaine hydrochloride. FLE is the liquid crystal precursor system with 5% lidocaine hydrochloride and 0.001% epinephrine. FC is the commercial formulation of lidocaine 50 mg/g EMS orange flavor dermatological ointment.

Sample	Group	Group Score	Sample Toxicity Classification
Positive control	1	14	Irritant
	2	16	Irritant
	3	15	Irritant
Negative control	1	0	Non-irritant
	2	0	Non-irritant
	3	0	Non-irritant
F	1	7	Moderate irritant
	2	6	Moderate irritant
	3	6	Moderate irritant
FL	1	6	Moderate irritant
	2	5	Slight irritant
	3	6	Moderate irritant
FLE	1	5	Slight irritant
	2	5	Slight irritant
	3	3	Slight irritant
FC	1	7	Moderate irritant
	2	6	Moderate irritant
	3	7	Moderate irritant

**Table 5 pharmaceuticals-18-01166-t005:** Classification of substances based on the endpoint assessment method following Protocol No. 96, as published by the European Centre for the Validationn of Alternative Methods [27,28] (*n* = 18), with representative images for each irritancy category.

Score	Classification	Representative Image
-	Non-irritant	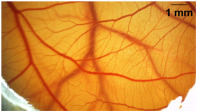
<6	Slight irritant	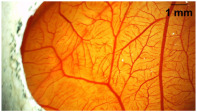
6 ≤ S ≤ 12	Moderate irritant	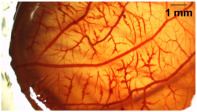
12 ≤ S ≤ 16	Irritant	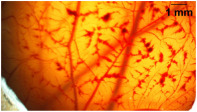
≥16	Severe irritant	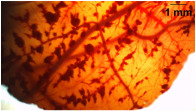

Note: The representative images included were selected from a reference panel routinely used in our laboratory, based on typical reactions observed in HET-CAM assays.

## Data Availability

The original contributions presented in the study are included in the article and Appendix A, further inquiries can be directed to the corresponding author.

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
