# Peer review of "In Vitro Proof-of-Concept Study: Lidocaine and Epinephrine Co-Loaded in a Mucoadhesive Liquid Crystal Precursor System for Topical Oral Anesthesia"

_pharmaceuticals, 2025, doi:10.3390/ph18081166_

Round 1
Reviewer 1 Report
Comments and Suggestions for Authors
The manuscript presents a novel and promising formulation for pain reduction during dental procedures using a lidocaine-epinephrine liquid crystalline precursor system (LCPS). The study is well-structured, methodologically sound, and scientifically significant in the field of dental biomaterials and drug delivery. The authors provide compelling evidence supporting the physicochemical, rheological, and mucoadhesive advantages of their formulation compared to a commercial topical anesthetic. The comment to revise the work are given below
- Please add literature or regulatory reference to use combinator of lidocaine and epinephrine.
- Improve introduction writing as its very scattered, improve flow and merge small paragraphs.
- Line 67: How Liquid crystalline precursor systems (LCPS) reduce side effects?
- Line 71-74: Additionally, these systems improve bioavailability by en-71 hancing the solubility of poorly soluble drugs, ensuring that a higher propor-72 tion of the active ingredient reaches the target site in the body, thereby increas-73 ing the overall effectiveness of the treatment. Is this relevant here? As this work/system is for topical.
- Figure 7: Right and left few images are same, why?
- Why, PPG-5-CETETH-20 used?
- SAXS peak assignments and d-spacing values should be quantitatively presented to support lamellar phase confirmation.
- The exact composition of the precursor (percentages of surfactant, oil, water phase) should be clearly tabulated.
- Details regarding the instrument settings and mucoadhesive testing protocol (e.g., model membrane, measurement method) should be elaborated for reproducibility.
- It would be useful to present cumulative permeation profiles over time, and flux (J) and lag time values to better understand release kinetics.
- Consider including drug release mechanism modeling (e.g., Higuchi, Korsmeyer-Peppas) to interpret the diffusion profile.
- Clear identification of the cell line or model system used, assay details (e.g., MTT, LDH), exposure time, and concentrations tested.
- If cytotoxicity assays were not conducted on oral epithelial or gingival cell lines, this should be justified.
- Epinephrine is notoriously unstable in aqueous formulations. Were any antioxidants or stabilizers used in the LCPS? Data on its chemical stability over time would enhance the manuscript.
Author Response
Reviewer #1:
The manuscript presents a novel and promising formulation for pain reduction during dental procedures using a lidocaine-epinephrine liquid crystalline precursor system (LCPS). The study is well-structured, methodologically sound, and scientifically significant in the field of dental biomaterials and drug delivery. The authors provide compelling evidence supporting the physicochemical, rheological, and mucoadhesive advantages of their formulation compared to a commercial topical anesthetic.
Thank you for your detailed and constructive feedback. Please find below our point-by-point responses and the corresponding revisions made to the manuscript.
The comment to revise the work are given below
- Please add literature or regulatory reference to use combinator of lidocaine and epinephrine.
The text has been revised to include regulatory and scientific support for the combined use of lidocaine and epinephrine. This combination is widely used in clinical practice due to its synergistic effects, where epinephrine promotes vasoconstriction, thereby prolonging the anesthetic effect of lidocaine and reducing its systemic absorption.
This practice is endorsed by major regulatory agencies:
- FDA (USA): approves injectable formulations containing lidocaine with epinephrine for dental use, as described in the Xylocaine® prescribing information (FDA, 2024).
- EMA (Europe): recognizes fixed-dose combinations of drugs, including local anesthetics, under the Guideline on Clinical Development of Fixed Combination Medicinal Products (EMA/CHMP/158268/2017).
- ANVISA (Brazil): registers and approves several commercial formulations containing lidocaine and epinephrine (e.g., Xylestesin®, Alphacaine®), and includes this combination in the RENAME 2024 (National List of Essential Medicines), recognizing it as essential for dental anesthesia. The formulation is approved for nerve blocks, local infiltration, extractions, and endodontic and periodontal procedures, and is distributed through public health units with funding from the SUS (Brazilian Unified Health System).
Additionally, the revised text cites both foundational and recent scientific literature supporting the efficacy of this combination:
- Becker & Reed (2012) Becker, D.E.; Reed, K.L. Essentials of local anesthetic pharmacology. Anesth. Prog. 2012, 59, 90–101. https://doi.org/10.2344/0003-3006-59.2.90
- Kalra et al. (2024) Kalra, G.; Makkar, S.; Menrai, N.; Kalia, V.; Suri, N.; Gupta, S. Comparative Evaluation of the Efficacy and Onset of Local Anesthesia Using Buffered 2% Lidocaine with 1:100,000 Adrenaline, Non-buffered 2% Lidocaine with 1:100,000 Adrenaline, Buffered 4% Articaine with 1:100,000 Adrenaline and Non-buffered 4% Articaine with 1:100,000 Adrenaline in Dental Extraction. J. Maxillofac. Oral Surg. 2024, 23, 1255–1260. https://doi.org/10.1007/s12663-023-01945-0
Inserted text in manuscript (Introduction, page 2):
The combination of lidocaine and epinephrine is widely used in clinical practice due to its synergistic effects, where epinephrine prolongs the anesthetic effect of lidocaine by vasoconstriction, reducing systemic absorption. This combination is endorsed by regulatory agencies such as the FDA, EMA, and RENAME [1-3] and is supported by both foundational pharmacological reviews and recent clinical studies demonstrating its efficacy in dental procedures [5,6].
References:
- FDA (2024), U.S. Food and Drug Administration. Xylocaine® (lidocaine hydrochloride and epinephrine) injection, USP – Prescribing Information. Revised April 2024. Available online:
- EMA (2017), European Medicines Agency (EMA). Guideline on Clinical Development of Fixed Combination Medicinal Products. EMA/CHMP/158268/2017. Available online: https://www.ema.europa.eu/en/documents/scientific-guideline/guideline-clinical-development-fixed-combination-medicinal-products-revision-2_en.pdf
- Ministério da Saúde (Brasil). Relação Nacional de Medicamentos Essenciais: RENAME 2024. Available online: https://bvsms.saude.gov.br/bvs/publicacoes/relacao_nacional_medicamentos_2024.pdf (accessed on 15 July 2025).
- Becker & Reed (2012), Becker, D.E.; Reed, K.L. Essentials of local anesthetic pharmacology. Anesth. Prog. 2012, 59, 90–101. https://doi.org/10.2344/0003-3006-59.2.90
- Kalra et al. (2024), Kalra, G.; Makkar, S.; Menrai, N.; Kalia, V.; Suri, N.; Gupta, S. Comparative Evaluation of the Efficacy and Onset of Local Anesthesia Using Buffered 2% Lidocaine with 1:100,000 Adrenaline, Non-buffered 2% Lidocaine with 1:100,000 Adrenaline, Buffered 4% Articaine with 1:100,000 Adrenaline and Non-buffered 4% Articaine with 1:100,000 Adrenaline in Dental Extraction. J. Maxillofac. Oral Surg. 2024, 23, 1255–1260. https://doi.org/10.1007/s12663-023-01945-0
- Improve introduction writing as its very scattered, improve flow and merge small paragraphs.
The introduction has been thoroughly revised to enhance logical flow and coherence. Smaller paragraphs have been merged, transitions between ideas have been clarified, and sentence structures have been refined to improve readability. Additionally, all changes have been highlighted in red in the revised manuscript to facilitate easy identification.
- Line 67: How Liquid crystalline precursor systems (LCPS) reduce side effects?
This reduction in side effects occurs because LCPS limit systemic absorption by maintaining the drug at the application site for longer periods. Their mucoadhesive properties also enhance retention on the mucosal surface, allowing for lower drug concentrations to be used effectively, which minimizes exposure to non-target tissues and reduces the risk of systemic toxicity. This information, along with the corresponding reference, has been included in the revised manuscript (lines 75-80).
- Line 71-74: Additionally, these systems improve bioavailability by enhancing the solubility of poorly soluble drugs, ensuring that a higher proportion of the active ingredient reaches the target site in the body, thereby increasing the overall effectiveness of the treatment. Is this relevant here? As this work/system is for topical.
Yes, this point is relevant, especially in the context of topical delivery. To clarify this relevance, we revised the sentence to emphasize the benefit of enhanced solubility in improving local drug concentration and therapeutic efficacy at the site of application.
Revised text in manuscript (page 3, lines 83-86):
These systems enhance the solubility of poorly soluble drugs, which is particularly beneficial for topical formulations, as it ensures higher local drug concentration at the mucosal surface, improving therapeutic efficacy.
- Figure 7: Right and left few images are same, why?
The images on the left (T0) and right (T5) sides of Figure 7 are not identical. They correspond to two different time points: T0 (before application) and T5 (after 5 minutes of exposure), for each experimental condition. In some cases, where the treatment caused no observable effect on the chorioallantoic membrane (CAM), the vascular pattern appears very similar at both time points. This visual similarity reflects the absence of vascular damage or irritation, supporting the biocompatibility of the tested formulation.
- Why, PPG-5-CETETH-20 used?
We selected PPG-5-CETETH-20 (Procetyl® AWS) due to its amphiphilic nature, biocompatibility, and well-documented ability to promote the formation of stable liquid crystalline phases, which contribute to both mucoadhesive properties and controlled drug release, key features for effective topical delivery in the oral mucosa. These characteristics made it a suitable component for our formulation strategy.
A detailed justification has now been added to the manuscript in Section 3 – Discussion, line 584-588, based on the original rationale already used during formulation development and previously cited. Therefore, no additional reference was required, and we maintained the existing citation in the revised text.
Procetyl® AWS (PPG-5-CETETH-20) has an amphiphilic structure, which enables the formation of stable liquid crystalline phases upon contact with aqueous environments. It is biocompatible and enhances both drug solubilization and mucoadhesion, making it particularly suitable for mucosal applications [33].
- SAXS peak assignments and d-spacing values should be quantitatively presented to support lamellar phase confirmation.
As requested, we have carefully considered the SAXS peak assignments and d-spacing values to support the identification of lamellar phases. These quantitative data are presented in Table S1 (Supplementary Material), where formulations such as F30, FLE30, and FLE100 exhibit q values in the ratios of 1:2:3, consistent with lamellar periodicity. This harmonic pattern is a hallmark of a lamellar structure, where Bragg reflections occur at integer multiples of the fundamental scattering vector (q, 2q, 3q). According to the equation , this corresponds to interplanar distances in the ratios d, d/2, and d/3, which is consistent with the periodicity of parallel layers. These quantitative data, along with the structural classification, confirm the presence of the lamellar phase in these formulations.
Regarding FL100, although only two peaks were observed (q and 2q), the d-spacing ratio (d1/d2 = 2) suggests a lamellar arrangement. However, as discussed in the manuscript, we acknowledge the complexity of liquid crystalline systems and the potential for phase coexistence or transitions. This is reflected in the PLM classification of FL100 as cubic.
To address this point, we have revised the manuscript to clarify the interpretation of the SAXS data, particularly for FL100, and to emphasize the importance of using complementary techniques such as SAXS and PLM for accurate structural characterization. The updated paragraph can be found in the Results and Discussion section (lines 214–228, page 6).
Further exploration revealed that F30, FLE30, and FLE100 displayed three SAXS peaks with q values in the ratios of 1:2:3, characteristic of lamellar mesophases. FL100, although exhibiting only two peaks with a d₁/d₂ ratio of 2, also suggests a lamellar arrangement. In PLM, F30 and FLE30 exhibited Maltese crosses, confirming the presence of lamellar phases. In contrast, FL100 and FLE100 were classified as cubic under PLM, despite their SAXS profiles suggesting lamellar characteristics. This divergence may indicate transitional or coexisting mesophases. The presence of striated textures or dark fields in PLM further supports the notion that systems with liquid-crystalline arrangements are structurally complex, requiring the integration of complementary techniques for accurate characterization. This interplay between SAXS and PLM highlights the nuanced behavior of these systems and underscores the importance of a multifaceted analytical approach, as also discussed by Stribeck in the context of lamellar two-phase systems with structural heterogeneity and stacking disorder (Stribeck, 1995).
- The exact composition of the precursor (percentages of surfactant, oil, water phase) should be clearly tabulated.
We thank the reviewer for the valuable suggestion. In response, we have included the exact composition of the precursor system—including the percentages of surfactant (PPG-5-CETETH-20 - Procetyl), oil (oleic acid), and aqueous phase (0.5% chitosan dispersion)—in a clearly organized table. This information has been added on page 4, lines 159 to 162, and is also presented in Table S1 of the Supplementary Material.
Table S1 includes the compositions of the drug-loaded formulations (with lidocaine and epinephrine) and their respective dilutions with artificial saliva (30% and 100%), ensuring full transparency and reproducibility of the formulation process.
Table S1- Composition (%) of the liquid crystalline precursor systems (LCPS), active compounds (lidocaine and epinephrine) and saliva-diluted formulations.
|
Formulation |
PPG-5-CETETH-20 (%, w/w) |
Oleic Acid (%, w/w) |
0.5% (w/v) Chitosan Dispersion (%, w/w) |
Lidocaine (%, w/w) |
Epinephrine (%, w/w) |
Artificial Saliva Addition (%, w/w) |
|
F |
40 |
30 |
30 |
– |
– |
- |
|
F30 |
40 |
30 |
30 |
– |
– |
30 |
|
F100 |
40 |
30 |
30 |
– |
– |
100 |
|
FL |
40 |
30 |
30 |
5 |
– |
- |
|
FL30 |
40 |
30 |
30 |
5 |
– |
30 |
|
FL100 |
40 |
30 |
30 |
5 |
– |
100 |
|
FLE |
40 |
30 |
30 |
5 |
0.001 |
- |
|
FLE30 |
40 |
30 |
30 |
5 |
0.001 |
30 |
|
FLE100 |
40 |
30 |
30 |
5 |
0.001 |
100 |
|
FC |
– |
– |
– |
5 |
– |
- |
|
FC30 |
– |
– |
– |
5 |
– |
30 |
|
FC100 |
– |
– |
– |
5 |
– |
100 |
- Details regarding the instrument settings and mucoadhesive testing protocol (e.g., model membrane, measurement method) should be elaborated for reproducibility.
We have significantly expanded Section 4.5 of the manuscript to provide a detailed description of the mucoadhesive testing protocol (lines 935-947).
The revised section now includes:
- The origin, preparation, and handling of the porcine buccal mucosa used as the model membrane.
- A full description of the experimental setup, including the custom-designed A/MUC device and its configuration.
- All relevant instrument settings for the TA-XT Plus Texture Analyzer (Stable Micro Systems, UK), including load cell capacity, contact force, contact time, test speeds, and temperature.
- The parameters measured (peak mucoadhesive force and mucoadhesion work) and the software used for analysis (Exponent®).
- The number of replicates performed.
To further enhance clarity and reproducibility, we have added illustrative images of the experimental setup in the Supplementary Material (Figure S1A–D).
We believe these additions fully address the reviewer’s concern and significantly improve the methodological transparency of our study.
- It would be useful to present cumulative permeation profiles over time, and flux (J) and lag time values to better understand release kinetics.
The permeation profiles shown in Figure 6 already represent the cumulative amount of lidocaine permeated over time (µg/cm²). To ensure this is clear to readers, we included an explicit statement in the main text clarifying that the data reflect cumulative permeation.
In addition, the steady-state flux (Jss) and lag time were calculated from the linear portion of the permeation curves using Fick’s first law, as described in Section 4.7 – In vitro permeation studies, and are reported in Table 3. These parameters are statistically analyzed and discussed in relation to the structural and rheological characteristics of the formulations, supporting the interpretation of their release and permeation kinetics. We believe these clarifications fully address the reviewer’s request and enhance the understanding of the release and permeation kinetics of the tested formulations.
- Consider including drug release mechanism modeling (e.g., Higuchi, Korsmeyer-Peppas) to interpret the diffusion profile.
As described in Section 2.2 and summarized in Table 2 of the manuscript, we have already applied several kinetic models to interpret the drug release profiles, including Higuchi and Korsmeyer-Peppas, as well as zero-order, first-order, and Weibull models.
The Korsmeyer-Peppas model yielded an n value of 0.2765 for FC, indicating Fickian diffusion, and 0.5105 for FL, suggesting a borderline case between Fickian and anomalous transport. The Higuchi model also showed a strong fit for FL (R² = 0.9723), supporting a diffusion-controlled release mechanism. These findings are consistent with the Weibull model, which provided the best overall fit (R² > 0.99) and further confirmed a Fickian diffusion mechanism based on the shape parameter b < 1.
We have ensured that these interpretations are clearly presented in the manuscript to support the understanding of the drug release mechanisms.
- Clear identification of the cell line or model system used, assay details (e.g., MTT, LDH), exposure time, and concentrations tested.
In our study, we employed the Hen’s Egg Test on the Chorioallantoic Membrane (HET-CAM), following Protocol No. 96 from the European Centre for the Validation of Alternative Methods (ECVAM), as described in Section 4.8 of the manuscript. This model is a well-established and validated in vivo alternative method for assessing the irritation potential of chemical substances. Although it does not involve cell lines or colorimetric assays such as MTT or LDH, the HET-CAM test offers a reliable and ethically accepted approach to evaluate acute vascular reactions—namely, hemorrhage, lysis, and coagulation—which are critical endpoints for formulations intended for topical mucosal application.
- If cytotoxicity assays were not conducted on oral epithelial or gingival cell lines, this should be justified.
We confirm that cytotoxicity assays using oral epithelial or gingival cell lines were not conducted in this study. Instead, we selected the HET-CAM assay due to its regulatory acceptance, ethical advantages, and high biological relevance. This in vivo alternative model allows for the evaluation of acute vascular responses—such as hemorrhage, lysis, and coagulation—on a live, vascularized membrane, which provides a more physiologically integrated and predictive response than conventional in vitro assays with isolated cell cultures.
The HET-CAM test has been widely used in the safety assessment of formulations for mucosal application, including the oral cavity, and is recognized as a reliable alternative to animal testing. To address the reviewer’s concern, we have now included a justification for the use of the HET-CAM model in the Materials and Methods section (Section 4.8, page 27, lines 1049-1059) of the revised manuscript.
- Epinephrine is notoriously unstable in aqueous formulations. Were any antioxidants or stabilizers used in the LCPS? Data on its chemical stability over time would enhance the manuscript.
Thank you for your insightful comment. No antioxidants or stabilizers were used in the LCPS formulations. This decision was intentional and aligned with the experimental design: all formulations were freshly prepared immediately before use, under light-protected conditions, and were not stored for extended periods. During handling and application, the formulations were kept shielded from light to minimize potential degradation of epinephrine.
As widely documented in the literature, epinephrine is chemically unstable in aqueous environments, mainly due to its catechol structure, which is highly susceptible to oxidation. Upon exposure to light, oxygen, or alkaline pH, epinephrine degrades into adrenochrome and other byproducts, typically accompanied by a visible color change (from colorless to pink or brown). In our study, by limiting the time between preparation and use, we minimized the risk of significant degradation, making the use of stabilizers unnecessary within the context of this experimental setting. We have now included statements in the revised manuscript to clarify that no stabilizers were used due to the fresh preparation protocol. This information was added to Section 4 – Materials and Methods (page 24, item 4.2) and further discussed in Section 3 – Discussion (page 21), where we also highlight the importance of investigating stability strategies in future formulation development.
Reviewer 2 Report
Comments and Suggestions for Authors
The manuscript presents a promising new approach to topical oral anesthesia with a strong experimental foundation. However, before publication, the authors should address the points above to strengthen the scientific rigor, clarity, and translational relevance of their work.
- Provide more quantitative data and statistical analysis in the main text, particularly for rheological, mucoadhesive, and drug release studies.
- Expand on the toxicity assessment methods and discuss the limitations of in vitro findings.
- Clearly state the limitations of the study and outline a roadmap for future in vivo and clinical validation.
- Condense and clarify sections of the manuscript to improve readability.
- Consider including a table summarizing the key physicochemical and functional properties of the new formulation versus the commercial control.
- The toxicity assessment is described as showing "similar toxicity" to the commercial formulation, but the methods and endpoints used for toxicity testing are not detailed in the provided excerpt. These should be clarified, and any limitations of the in vitro toxicity model should be acknowledged
- There are several minor typographical and grammatical errors throughout the text. A thorough language edit is recommended.
- The work is limited to in vitro characterization, as indicated by the authors in the draft. While this is appropriate for a proof-of-concept, the lack of in vivo data means that claims regarding clinical safety and efficacy remain speculative. The authors should more clearly state these limitations and outline specific next steps for translation.
- The study reports a 93% reduction in drug permeation flow compared to the commercial formulation, but the clinical relevance of this finding is not fully discussed. It is unclear whether this reduction could compromise the onset or depth of anesthesia in vivo, or if it simply reflects improved retention at the application site.
- The study appropriately compares the new formulation to a commercial lidocaine ointment, highlighting differences in rheological behavior and microstructure upon dilution with artificial saliva. However, details regarding the reproducibility of the phase diagram and whether the selected formulation is optimal in terms of drug loading, stability, and scalability are limited and should be discussed.
Author Response
Reviewer #2:
The manuscript presents a promising new approach to topical oral anesthesia with a strong experimental foundation. However, before publication, the authors should address the points above to strengthen the scientific rigor, clarity, and translational relevance of their work.
We thank the reviewer for the positive evaluation of our work and for recognizing the potential of our approach to topical oral anesthesia. We have carefully addressed all the specific comments and suggestions provided, with the aim of improving the scientific rigor, clarity, and translational relevance of the manuscript. Detailed responses to each point are provided above, and the corresponding revisions have been incorporated into the manuscript.
- Provide more quantitative data and statistical analysis in the main text, particularly for rheological, mucoadhesive, and drug release studies.
We would like to clarify that the quantitative results and statistical analyses for the rheological, mucoadhesive, and drug release studies are already highlighted in the main text of the manuscript. These include mean values, standard deviations, and significance levels, along with appropriate graphical representations.
- Expand on the toxicity assessment methods and discuss the limitations of in vitro
We have expanded the discussion on the toxicity assessment by including new paragraphs in the Discussion section. This addition clarifies the rationale for using the HET-CAM assay, highlights its relevance for mucosal formulations, and addresses its limitations as an alternative in vivo model. We also emphasize the need for further in vivo studies in mammalian models to confirm the safety and biocompatibility of the developed formulations.
We hope this addition addresses your concern and strengthens the translational relevance of our findings.
- Clearly state the limitations of the study and outline a roadmap for future in vivo and clinical validation.
We have revised the Discussion section (line 758-767) to clearly state the limitations of the present study, particularly the absence of in vivo efficacy data. We also included a detailed roadmap for future validation, outlining the need for in vivo studies in appropriate animal models to assess anesthetic efficacy, local tolerability, and systemic absorption. Furthermore, we highlighted the importance of subsequent clinical trials, long-term stability studies, and scalability assessments to support clinical translation and regulatory approval. These additions aim to strengthen the translational relevance and scientific rigor of the manuscript.
- Condense and clarify sections of the manuscript to improve readability.
We carefully reviewed the manuscript and revised several sections to improve clarity and readability. We condensed overly detailed descriptions, removed redundant phrases, and restructured some paragraphs to enhance the logical flow of information. These changes were made while preserving the scientific content and integrity of the data. We believe these edits have significantly improved the overall readability and accessibility of the manuscript.
- Consider including a table summarizing the key physicochemical and functional properties of the new formulation versus the commercial control.
We have included a comparative summary Table highlighting the key physicochemical, rheological, and functional properties of the developed formulation (FLE) versus the commercial control (FC). This table includes parameters such as viscosity behavior, rheological profile, mechanical and mucoadhesive properties, drug release, permeation, and toxicity classification.
To maintain the clarity and flow of the main manuscript, we have included this table as Supplementary Table S5. A reference to this table has also been added in the Discussion section to guide readers to the comparative data (line 556-558).
- The toxicity assessment is described as showing "similar toxicity" to the commercial formulation, but the methods and endpoints used for toxicity testing are not detailed in the provided excerpt. These should be clarified, and any limitations of the in vitro toxicity model should be acknowledged
We appreciate the opportunity to clarify the toxicity assessment. In our study, the Hen’s Egg Test on the Chorioallantoic Membrane (HET-CAM) was employed, following Protocol No. 96 from the European Centre for the Validation of Alternative Methods (ECVAM). This assay evaluates the acute irritation potential based on three vascular endpoints: hemorrhage, lysis, and coagulation, each scored on a scale from 0 (no reaction) to 3 (severe reaction). The highest score observed among the three effects determines the final irritancy classification.
Although often described as an alternative method, we emphasize that the HET-CAM assay is an in vivo model, as it involves the use of a live embryonated egg with a fully vascularized extraembryonic membrane. It offers several advantages, including ethical acceptability (since the embryos are used before the onset of pain perception), low cost, and the ability to visually assess vascular responses in real time.
We acknowledge, however, that this model does not fully replicate the complexity of human mucosal tissues, lacks systemic metabolism and immune responses, and is limited to evaluating acute, rather than chronic, toxicity. Despite these limitations, it provides a reliable and predictive tool for the initial screening of mucosal irritation, particularly for formulations intended for topical use in the oral cavity.
To address your comment, we have now included a paragraph in the discussion section explicitly describing the HET-CAM method, its endpoints, advantages, and limitations, along with a justification for its selection in the context of this proof-of-concept study.
- There are several minor typographical and grammatical errors throughout the text. A thorough language edit is recommended.
The document has been reviewed and revised for grammar and typographical accuracy.
- The work is limited to in vitro characterization, as indicated by the authors in the draft. While this is appropriate for a proof-of-concept, the lack of in vivo data means that claims regarding clinical safety and efficacy remain speculative. The authors should more clearly state these limitations and outline specific next steps for translation.
We fully agree that the absence of in vivo data is a significant limitation of the current study. As noted in the Discussion section, this work was designed as a proof-of-concept, and the findings are preliminary. We have clearly acknowledged that clinical safety and efficacy of the developed formulations remain speculative at this stage. To address this, we have explicitly stated in the manuscript that further in vivo studies are necessary to evaluate the anesthetic performance and clinical applicability of the formulations. We have also outlined the next steps for translation, which include in vivo efficacy and safety assessments in appropriate animal models, followed by clinical trials. These additions aim to ensure that the conclusions are interpreted with appropriate caution and to provide a clear roadmap for future research.
- The study reports a 93% reduction in drug permeation flow compared to the commercial formulation, but the clinical relevance of this finding is not fully discussed. It is unclear whether this reduction could compromise the onset or depth of anesthesia in vivo, or if it simply reflects improved retention at the application site.
We agree that the clinical relevance of the observed 93% reduction in drug permeation flow is an important point. We would like to clarify that this aspect has already been addressed in the Discussion section of the manuscript (page 23, line 698-720). Specifically, we explain that although the total permeation was lower, the developed formulation exhibited immediate onset of permeation and a sustained release profile. These characteristics suggest that the formulation may still provide a rapid onset of anesthesia while enhancing local retention and prolonging the anesthetic effect. We also acknowledge that further in vivo studies are necessary to confirm these findings and fully assess clinical efficacy. To address the reviewer’s concern, we have revised the corresponding paragraph in the Discussion to make the clinical relevance of the in vitro permeation data more explicit.
- The study appropriately compares the new formulation to a commercial lidocaine ointment, highlighting differences in rheological behavior and microstructure upon dilution with artificial saliva. However, details regarding the reproducibility of the phase diagram and whether the selected formulation is optimal in terms of drug loading, stability, and scalability are limited and should be discussed.
We agree that additional discussion regarding the reproducibility of the phase diagram and the optimization of the selected formulation is important. In the revised manuscript, we have now included a paragraph in the Discussion section (page 20, line 557) addressing the reproducibility of the phase diagram, the rationale for selecting the final formulation based on its physicochemical and mucoadhesive properties, and considerations for drug loading, stability, and potential scalability. These aspects are critical for future translational development and have now been clarified accordingly.
Round 2
Reviewer 1 Report
Comments and Suggestions for Authors
Most of comment justified and revised appropriately.
1. Please improve the discussion of drug release and permeation, with addition of correlation to in vivo for better understanding. Please refer this work.
2. Does, Table 5 Data is visual observed? Add photographic images for better understanding. Refer similar work for better understanding. https://doi.org/10.2147/DDDT.S107917, https://doi.org/10.3389/fphar.2018.01461, https://doi.org/10.2147/IJN.S219670
Author Response
Dear Reviewer,
We appreciate the opportunity to revise our manuscript following the minor revisions suggested by you. We have carefully addressed the comments and updated the manuscript accordingly. All modifications have been highlighted using track changes for ease of review.
We would like to thank you and the editorial team for their valuable feedback and consideration. We look forward to the final decision.
With best regards,
Giovana
Reviewer 1:
Most of comment justified and revised appropriately.
We thank the reviewer for their feedback and are pleased that the revisions were considered appropriate.
- Please improve the discussion of drug release and permeation, with addition of correlation to in vivo for better understanding. Please refer this work.
We thank the reviewer for the suggestion. As requested, we have expanded the Discussion section to include a correlation between the in vitro drug permeation and the expected in vivo performance. To support this enhancement, we have incorporated the recommended reference (https://doi.org/10.2147/IJN.S219670), which discusses the pharmacokinetics and mitochondrial delivery of lidocaine formulations. This addition strengthens the translational relevance of our findings.
The corresponding revision appears on page 25, lines 845 to 867 of the updated manuscript.
- Does, Table 5 Data is visual observed? Add photographic images for better understanding. Refer similar work for better understanding. https://doi.org/10.2147/DDDT.S107917, https://doi.org/10.3389/fphar.2018.01461, https://doi.org/10.2147/IJN.S219670
We thank the reviewer for this important comment. As requested, we have updated Table 5 to include representative photographic images illustrating each irritancy category. These images are part of a reference panel routinely used in our laboratory for HET-CAM evaluations and were selected to exemplify typical visual endpoints (hemorrhage, lysis, and coagulation) associated with the scoring system described in Protocol No. 96 (DB-ALM). This visual enhancement is aimed at improving the reader’s understanding of the classification method applied.

Reviewer 2 Report
Comments and Suggestions for Authors
The authors have reflected all the said suggestions and comments, which made the manuscript enhanced with improved readability; Thus, I suggest for further consideration with acceptance.
Author Response
Reviewer #2:
The authors have reflected all the said suggestions and comments, which made the manuscript enhanced with improved readability; Thus, I suggest for further consideration with acceptance.
We sincerely thank the reviewer for their positive feedback and thoughtful evaluation. We are pleased that the revisions have contributed to improving the clarity and readability of the manuscript. We greatly appreciate the recommendation for acceptance.
